# Fluorescent sensors for imaging of interstitial calcium

Ariel A. Valiente-Gabioud [1], Inés Garteizgogeascoa Suñer [2,5],
Agata Idziak [3,5], Arne Fabritius[1], Jérome Basquin[4], Julie Angibaud[3],
U. Valentin Nägerl [3], Sumeet Pal Singh [2] & Oliver Griesbeck [1] ✉

Calcium in interstitial fluids is central to systemic physiology and a crucial ion pool for entry into cells through numerous plasma membrane channels. Its study has been limited by the scarcity of methods that allow monitoring in tight inter-cell spaces of living tissues. Here we present high performance ultra-low affinity genetically encoded calcium biosensors named GreenT-ECs. GreenT-ECs combine large fluorescence changes upon calcium binding and binding affinities (Kds) ranging from 0.8 mM to 2.9 mM, making them tuned to calcium concentrations in extracellular organismal fluids. We validated GreenT-ECs in rodent hippocampal neurons and transgenic zebrafish in vivo, where the sensors enabled monitoring homeostatic regulation of tissue interstitial calcium. GreenT-ECs may become useful for recording very large calcium transients and for imaging calcium homeostasis in inter-cell structures in live tissues and organisms.

Calcium acts as a crucial second messenger within cells, involved in regulating a plethora of cellular processes. The complexities of these cellular signaling mechanisms, their spatiotemporal dynamics and the many channels, buffers, pumps and downstream effector enzymes involved, have been extensively studied[1,2]. In contrast, calcium dynamics and regulation in interstitial fluids have been far less investigated. Free ionic calcium in these fluids forms a critical reservoir for entry into cells through several classes of plasma-membrane calcium channels. It is in equilibrium with bound calcium in proteins, extracellular matrix components and is present in high amounts in bone and teeth. Multicellular organisms have developed means to tightly control and regulate its concentrations in circulating fluids. A central role here is played by the Calcium Sensing Receptor (CaSR), a G-protein coupled receptor that, among other ligands, senses extracellular calcium and transduces intracellular signaling[3,4]. Its expression in the parathyroid gland, the secretion of parathyroid hormone by dedicated cells in the parathyroid gland, and the coupling to active vitamin D[5–7] signaling are key components of a central regulatory system that adjusts free

calcium in body fluids to a narrow concentration range[7]. While central mechanisms of calcium homeostasis in interstitial fluids appear well understood, less is known about its regulation in extended and narrow inter-cell spaces that may be finely spun and far away from known homeostatic regulatory centers. The use of ion-selective electrodes, magnetic nanoparticles and microdialysis has opened a window into complex dynamics and modes of its regulation in extracellular volumes[8,9], however, these techniques typically suffer from low spatiotemporal resolution. Imaging of fluorescent calcium indicators, in combination with advanced microscopy, offers excellent spatiotemporal resolution[10]. Genetically encoded calcium indicator variants[11] allow precise localization in tissues and at subcellular sites of interest. Unfortunately, the focus of sensor development lay almost exclusively on applications within cells or cellular organelles, resulting in excellent sensors with high (nM) affinities for calcium. Sensors with intermediate to low affinity were engineered to image calcium in cellular organelles where free calcium concentrations may reach levels of several hundred micromolar[12–15]. Few efforts have been directed at developing sensors

[1]Max Planck Institute for Biological Intelligence, Tools for Bio-Imaging, Am Klopferspitz 18, 82152 Martinsried, Germany. [2]Institute de Recherche Interdisciplinaire en Biologie Humaine et Moléculaire (IRIBHM), 808 Route de Lennik, Université Libre de Bruxelles (ULB), 1070 Brussels, Belgium. [3]Institut Interdisciplinaire de Neurosciences, Synaptic Plasticity and Super-Resolution Microscopy, CNRS - Université de Bordeaux – 146 rue Léo-Saignat, Bordeaux, France. [4]Structural Cell Biology, Max-Planck-Institute for Biochemistry, Am Klopferspitz 18, Martinsried 82152, Germany. [5]These authors contributed equally: Inés Garteizgogeascoa Suñer, Agata Idziak. ✉e-mail: oliver.griesbeck@bi.mpg.de

with even lower affinities[16]. Here we present genetically encoded calcium biosensors with large fractional fluorescence changes and millimolar affinities to match concentrations found in various interstitial body fluids. We validate them in vitro and in several physiological assays.

## Results

### Development of GreenT-ECs

For imaging calcium in body fluids and intercell spaces, we engineered a set of low-affinity genetically encoded calcium sensors (Fig. 1). Our design was inspired by the molecular architecture of Camgaroo[17,18].

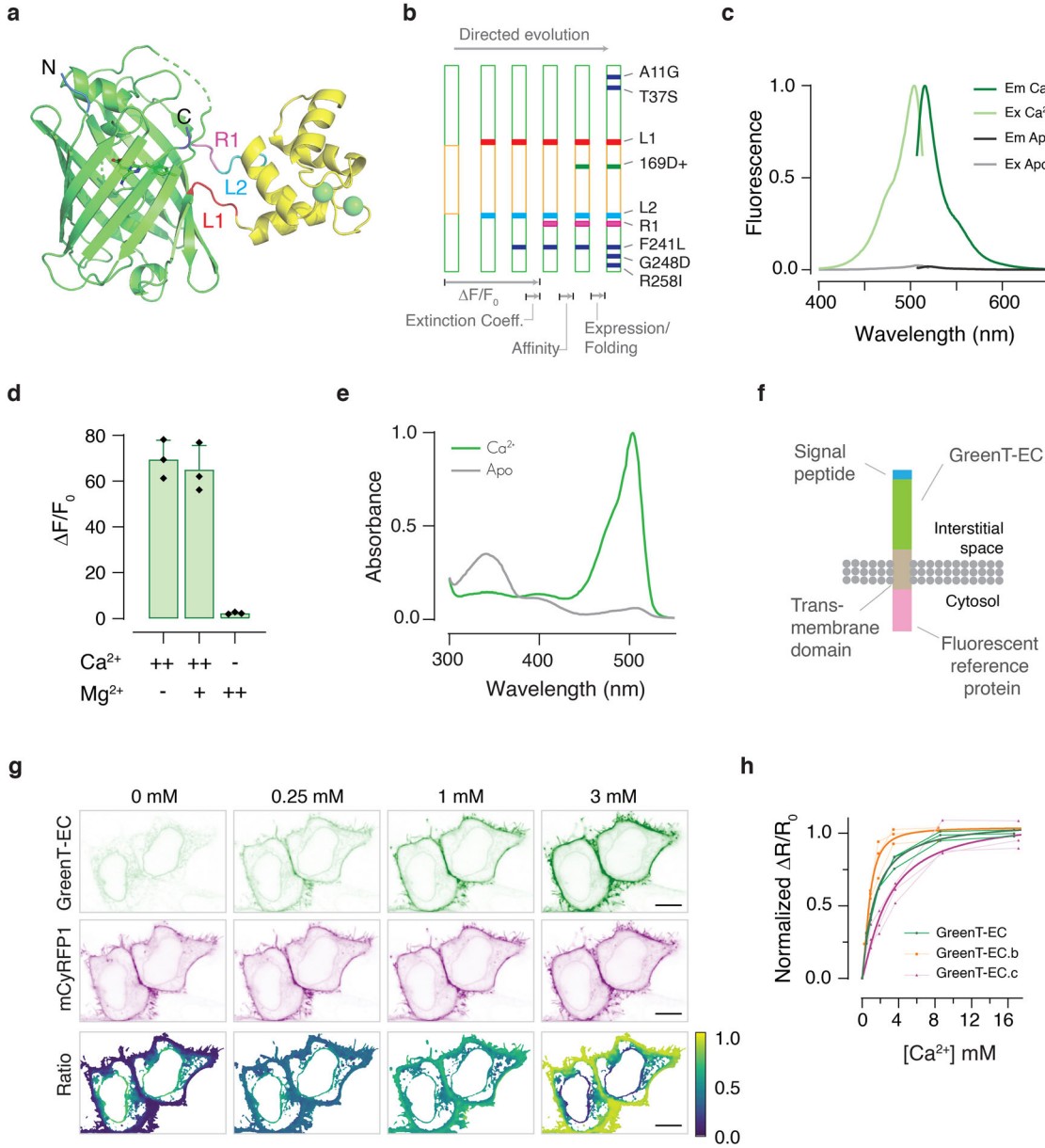

**Fig. 1 | Development of GreenT-ECs. a** Crystal structure of a calcium-bound GreenT-EC intermediate variant (NRS 1.2) generated during the process of directed evolution. The minimal calcium-binding domain derived from Troponin C (yellow) was inserted into the fluorescent protein mNeonGreen (green). Spheres (green) indicate two bound calcium ions. Linkers (L1 and L2), as well as a particularly crucial region (R1) for engineering are highlighted. N and C mark the N-and C-terminus of the protein. **b** Iterative cycles of directed evolution were performed to optimize GreenT-ECs (green: mNeonGreen moiety, yellow: TnC minimal domain). L1, L2: mutated linker amino acids. R1: stretch of three amino acids comprising residues 225–227 reaching into the ß-barrel. Together with linker mutations, this sensitive area fundamentally affected fluorescence change, brightness and off-kinetic of variants. Individual amino acid changes affecting folding and expression are indicated (blue). **c** Excitation and emission spectrum of GreenT-EC at 0 mM (gray) and 30 mM (green) $Ca^{2+}$. **d** Maximal fluorescence change of GreenT-EC in response to calcium and/or magnesium (++ is 100 mM, + is 5 mM, - is 0 mM, $n = 3$ technical replicates for one protein production).

**e** Absorbance changes of GreenT-ECs in the absence (grey) and presence (green) of calcium. The major absorbance peak maximum was at 504 nm. Minor absorbance peaks at 350 nm and 400 nm corresponded to dark forms of the chromophore. **f** Cell surface display of GreenT-EC. An effective way consisted of fusing an N-terminal signal peptide to GreenT-EC and a C-terminal membrane anchoring domain (from PDGF receptor). The fluorescent reference protein mCyRFP1 was added to the C-terminus (intracellular). **g** Representative images of HeLa cells expressing surface-delivered GreenT-ECs when exposed to increasing calcium concentrations. Ratiometric (GreenT-EC/mCyRFP1) images are presented in the bottom panel. Scale bar, 10 μm. **h** GreenT-EC/mCyRFP1 titrations on the surface of HEK293T cells. Individual titration experiments are plotted with lines and symbols ($n = 3$ biological replicates, with a total number of analyzed cells of 30, 29 and 26 for GreenT-EC, GreenT-EC.b and GreenT-EC.c, respectively). Fitted curves for each variant are included with thicker lines. Source data are provided in the Source Data file.

As calcium sensing moiety, we employed a minimal calcium binding domain derived from Troponin C (TnC)[19] and used it to replace Tryptophan 148 within the bright green fluorescent protein mNeonGreen[20]. A similar design for a high-affinity sensor had been presented earlier[21]. Compared to the GCaMP design[22,23], prominent in high-affinity biosensors, our design yielded sensors with fewer calcium binding sites per indicator (2 vs 4) and conserved original N-and C-termini of mNeonGreen for the fusion of subcellular targeting motifs. Due to the bright green fluorescence in the bound state, the insertion of the TnC fragment and the tuning of affinities for extracellular applications, we named the sensors GreenT-ECs. To elucidate more details on sensor function and for guidance for our protein engineering efforts, we solved a high-resolution crystal structure of an early variant of the sensor in the calcium-bound state to 1.3 A° resolution (Fig. 1a, Supplementary Table 1 and Supplementary Note). It revealed a structure in which the ß-barrel of mNeonGreen is opened to harbor the minimal calcium-binding domain of TnC. Within the TnC domain, the two EF-hand loops with two bound calcium ions and key regions linking the TnC domain to mNeonGreen are visible. Using structure-guided optimization and extended iterative directed evolution (Fig. 1b and Supplementary Note) we were able to dramatically enhance fluorescence changes and response behavior of the sensors. Fluorescence changes were finally about 60-fold in response to calcium binding and basal fluorescence in the absence of calcium was close to detection limits (Fig. 1c) (Table 1). The excitation and emission maxima were 504 nm and 515 nm, respectively. There was no detectable interference by magnesium (Fig. 1d). The binding affinity was drastically reduced by inserting either an additional aspartic acid after position 168 (GreenT-EC) or a serine after position 204 (GreenT-EC.b) within the EF-hand calcium chelating loops. Additionally, GreenT-EC.b, bearing the extra mutation T261I (GreenT-EC.c) presented the lowest calcium-binding affinity (Supplementary Fig. 2, Table 1). Altogether, these mutations allowed tuning the sensors to the low affinities necessary for use in interstitial fluids, without compromising the large fluorescence changes achieved. In vitro affinity titrations of purified recombinant proteins revealed sensors with Kds in the millimolar range. A comprehensive in vitro analysis of the spectroscopic properties of the sensors is available in Fig. 2. GreenT-ECs could be localized to the surface of mammalian cells by fusion of a number of appropriate targeting motifs (Fig. 3). The fluorescence properties of GreenT-ECs facilitated selection of optimal targeting motifs, as the indicators, when expressed in the cytosol, are almost nonfluorescent (Supplementary Fig. 3, 4), while brightly fluorescent when exposed to millimolar calcium concentrations found in extracellular liquids or buffers. We assayed surface exposure quantitatively using flow cytometry of transfected HeLa cells in the presence or absence of calcium in the buffer and identified favorable combinations of export signal peptides and transmembrane domains (Fig. 3). This allowed achieving a strong signal contrast between the bright fluorescence of surface exposed indicators versus the dim fluorescence of residual indicators inside cells within the cytosol or en route to the plasma membrane. Conceivably, dynamic probing of surface exposure or secretion of proteins might be an interesting secondary use of GreenT-ECs. A reference protein could be fused to the cytosolic C-terminal end of the sensor, facilitating calibration measurements. Using microscopy and flow cytometry, we assayed several spectrally compatible fluorescent proteins for enabling referencing while not compromising the surface exposure of the probe (Supplementary Fig. 5, 6). The red fluorescent protein mCyRFP1[24] was a useful candidate in this respect. It also allowed convenient co-excitation of the two fluorophores with a single laser wavelength in one photon excitation mode. Referenced GreenT-ECs could be titrated on the surface of HEK293 cells by sequentially increasing the free calcium concentration in extracellular buffers (Fig. 1h). These titrations yielded higher observed affinities than the measurements performed with purified recombinant protein. The Kds

**Table 1 | Spectroscopic properties of GreenT-ECs**

| In vitro properties | | | | | | | | | | | Cell surface titration | | |
|---|---|---|---|---|---|---|---|---|---|---|---|---|---|
| Variant | $\Delta F/F_O$ | EC | QY | QY*EC | kd (mM) | nH | $K_{off}$ (s-1) | $K_{obs}1$ (s-1) | $K_{obs}2$ (s-1) | pKa | $\Delta R/R_O$ | kd (mM) | nH |
| GreenT-EC | 61 | 79 | 0.72 | 57 | 7.1 ± 1.2 | 0.8 ± 0.1 | 33 | 0.35 (25%) | 0.05 | 6.7 | 10 ± 3 | 1.3 ± 0.2 | 1.1 ± 0.2 |
| GreenT-EC.b | 58 | 81 | 0.74 | 60 | 1.6 ± 0.1 | 1.5 ± 0.1 | 50 | 0.25 (25%) | 0.05 | 6.7 | 8 ± 2 | 0.8 ± 0.1 | 1.8 ± 0.6 |
| GreenT-EC.c | 56 | 80 | 0.72 | 58 | 20 ± 12 | 1.0 ± 0.2 | 50 | 0.12 (40%) | 0.04 | 6.7 | 10 ± 2 | 2.9 ± 1.9 | 1.2 ± 0.6 |

$\Delta F/F_O$, fluorescence change upon calcium binding; EC (mM$^{-1}$ cm$^{-1}$), extinction coefficient of the bound state; QY, quantum yield of the bound state; $n_H$, Hill coefficient; Kd, dissociation constant; $K_{off}$, off-rate kinetics; $K_{obs}1$ and $K_{obs}2$, the observed on-rate kinetics of the two-phase association process calculated at calcium saturation (the percentage of the response obtained during the fast phase is indicated in brackets). The errors in the Kd and $n_H$ values were obtained from a Specific binding with Hill coefficient fitting (GraphPad, Prism). A summary of the parameters obtained in the cell surface titration displayed in Fig. 1h is also included (left table). $\Delta R/R_O$, ratio change on HEK293T cells using mCyRFP1-referenced sensors presented in Fig. 1h. Three dishes were analyzed ($n = 3$ biological replicates) for each construct and the response of 8–10 cells were averaged for each dish (the full table is available in the Source Data file). The error in $\Delta R/R_O$ corresponds to the standard deviation.

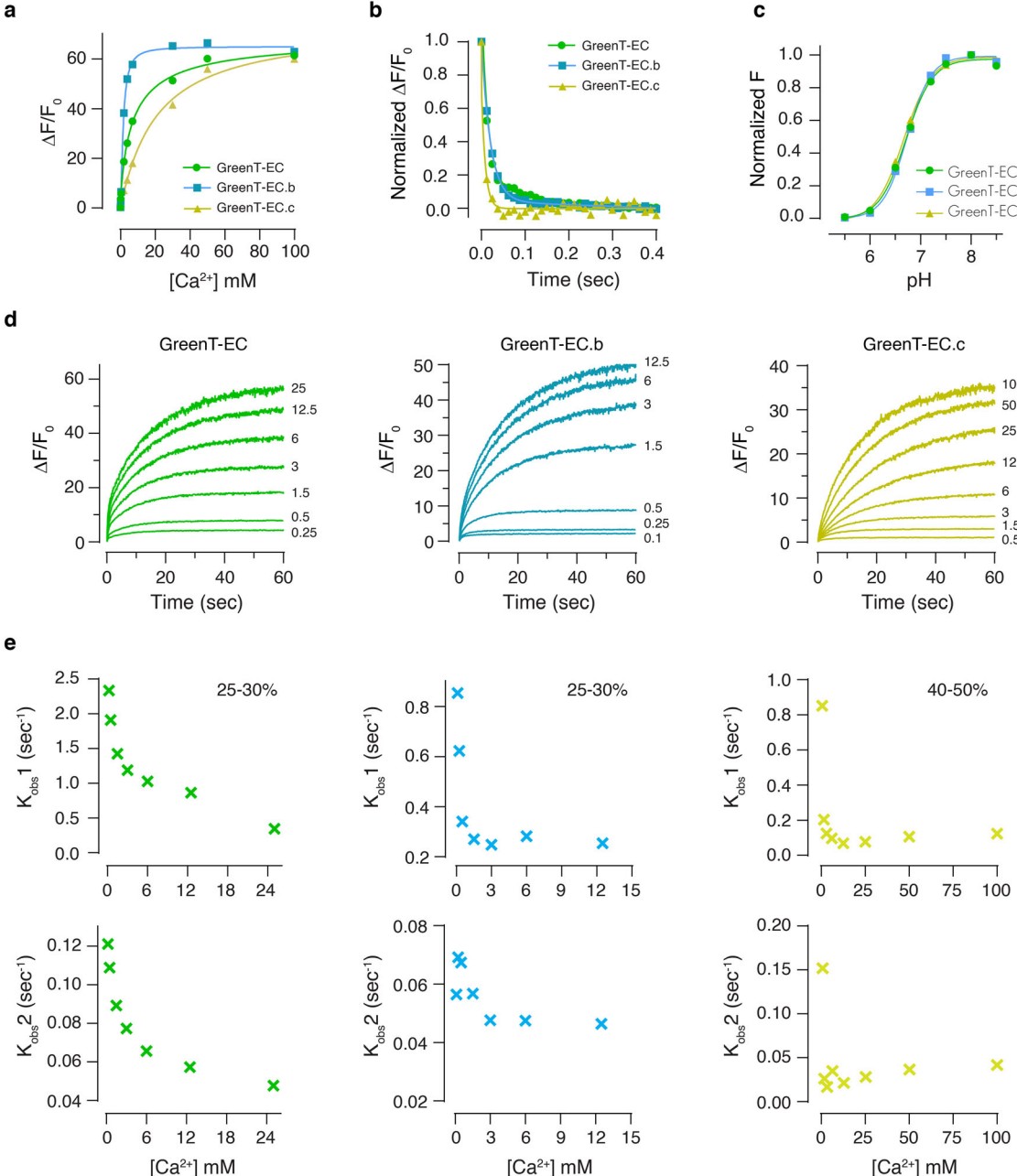

**Fig. 2 | In vitro spectroscopy of GreenT-ECs. a** In vitro affinity titration fitted curves of recombinant purified GreenT-EC and two variants with higher (GreenT-EC.b) and lower (GreenT-EC.c) affinity. **b** Off-rate kinetics of the three variants of the sensor. The proteins were prepared in MOPS 30 mM, KCl 100 mM, Mg 2.5 mM, Ca²⁺ 100 mM and rapidly mixed with BAPTA 200 mM. **c** Fitted curves of pka measurements for the three sensors studied in this article. Measurements were obtained by preparing pH solutions in MOPS/MES containing 100 mM Ca²⁺. **d** On-rate kinetics of GreenT-EC (green), GreenT-EC.b (blue) and lower affinity GreenT-EC.c (yellow) were measured by rapidly mixing solutions of protein in MOPS 30 mM, KCl 100 mM, Mg 1 mM with MOPS 30 mM, KCl 100 mM, Mg²⁺ 1 mM containing increasing concentration of Ca²⁺.

**e** A double exponential was used for the fittings for each variant and the obtained Kon values were plotted against the calcium concentration. $K_{obs}1$ corresponds to the first step of the rate, representing 25–30% of the total response. $K_{obs}2$ corresponds to the slower process and accounts for the higher percentage of the total response. In all cases, a decrease in the $K_{obs}$ is observed at increasing concentrations of calcium. This suggests a mechanism by which there is an equilibrium between at least two species of the sensor, and only one can progress to a fluorescent species upon calcium binding. Technical replicates of proteins produced and purified on the same day were used for all fittings. Source data are provided in the Source Data file.

for calcium were 1.3 mM (GreenT-EC), 0.8 mM (GreenT-EC.b) and 2.9 mM (GreenT-EC.c) (Fig. 1g, h) (Table 1).

**Validation of GreenT-EC in rodent hippocampus**

We next validated sensors in cultured rat hippocampal neurons (Supplementary Fig. 7) and mouse hippocampal slices (Fig. 4). For these experiments, we anchored GreenT-EC to the neuronal surface with a GPI-anchor (Fig. 4a)[25]. Both in transfected hippocampal neurons in

culture (Supplementary Fig. 7b) and in organotypic slices after AAV (Adeno-Associated Virus) delivery of GreenT-EC (Fig. 4b) we saw bright and specific surface labeling of transfected cells using confocal microscopy and super-resolution STED (stimulated emission depletion). Fluorescence was largely resistant to photobleaching, allowing for repetitive imaging of the same cells (Supplementary Fig. 8a). Moreover, fluorescence intensity dynamically changed when perfusing neurons with buffers containing 0 mM, 1.5 mM and 8 mM free calcium

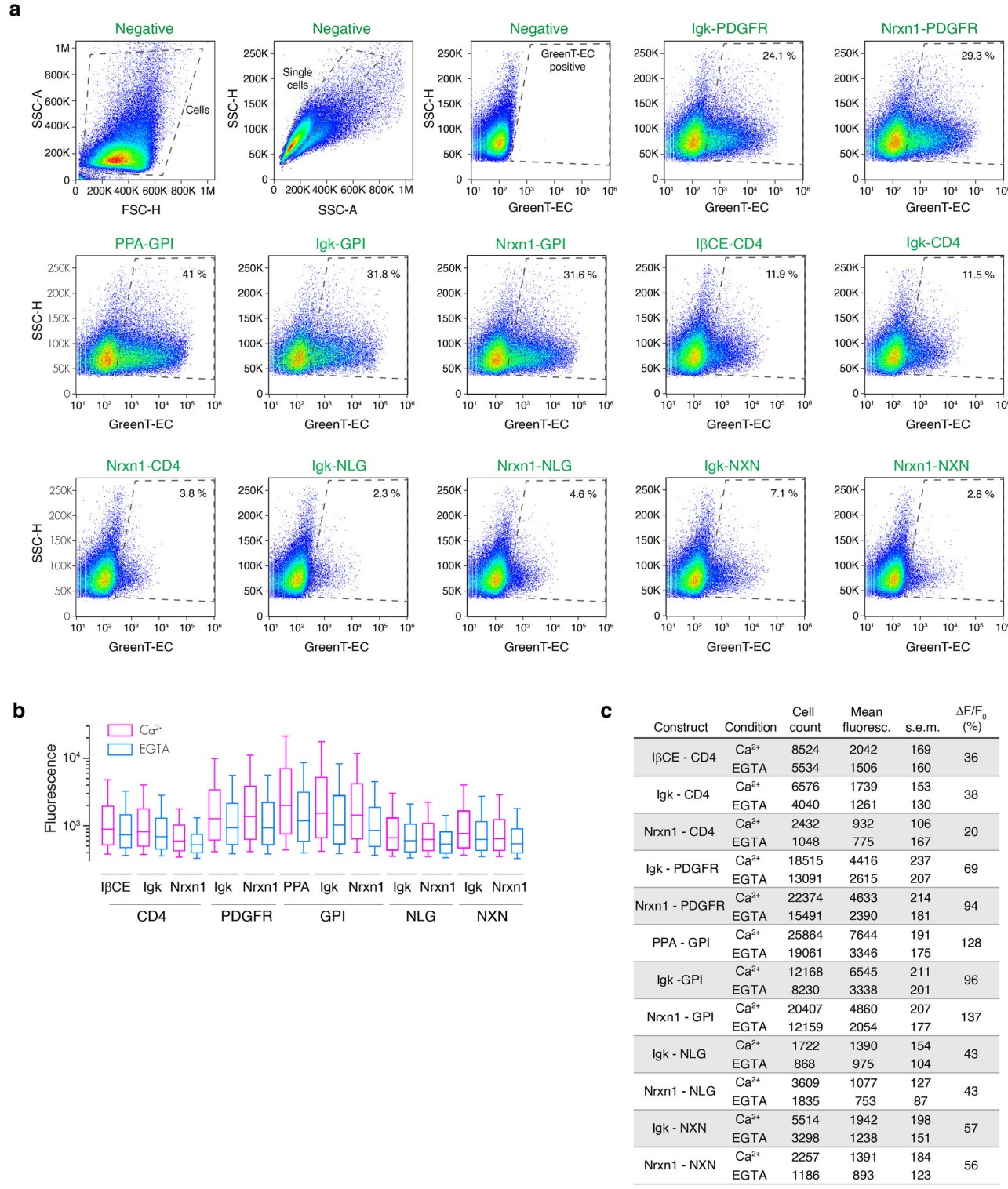

**Fig. 3 | Flow cytometry analysis of GreenT-EC cell surface exposure.** Twelve combinations of export signal peptides and transmembrane/anchoring domains were used to target GreenT-EC to the surface of the cell. HeLa cells were transiently transfected with each construct and analyzed using flow cytometry. **a** Density plots of the negative control (non-transfected) indicating the gating strategy for selecting GreenT-EC positive single cells (indicated in slashed lines). Both the negative and the rest of the constructs displayed were resuspended in a MOPS buffer containing 3 mM CaCl$_2$. **b** The fluorescence of GreenT-EC positive gated cells was further analyzed before and after the addition of EGTA (final concentration of 7.5 mM). Box-plots 90-10% are presented (whiskers are 90 and 10 percentiles, boxes are 75 and 25 percentiles and the center line correspond to the median). **c** Summary table compiling the main parameters of the experiment: Number of cells gated as GreenT-EC positive in each condition (n=cell count), mean fluorescence of GreenT-EC (mean fluoresce.), standard error of the mean (s.e.m) and the response calculated between the mean GreenT-EC fluorescence values before and after the addition of EGTA ($\Delta F/F_0$). Source data are provided in the Source Data file.

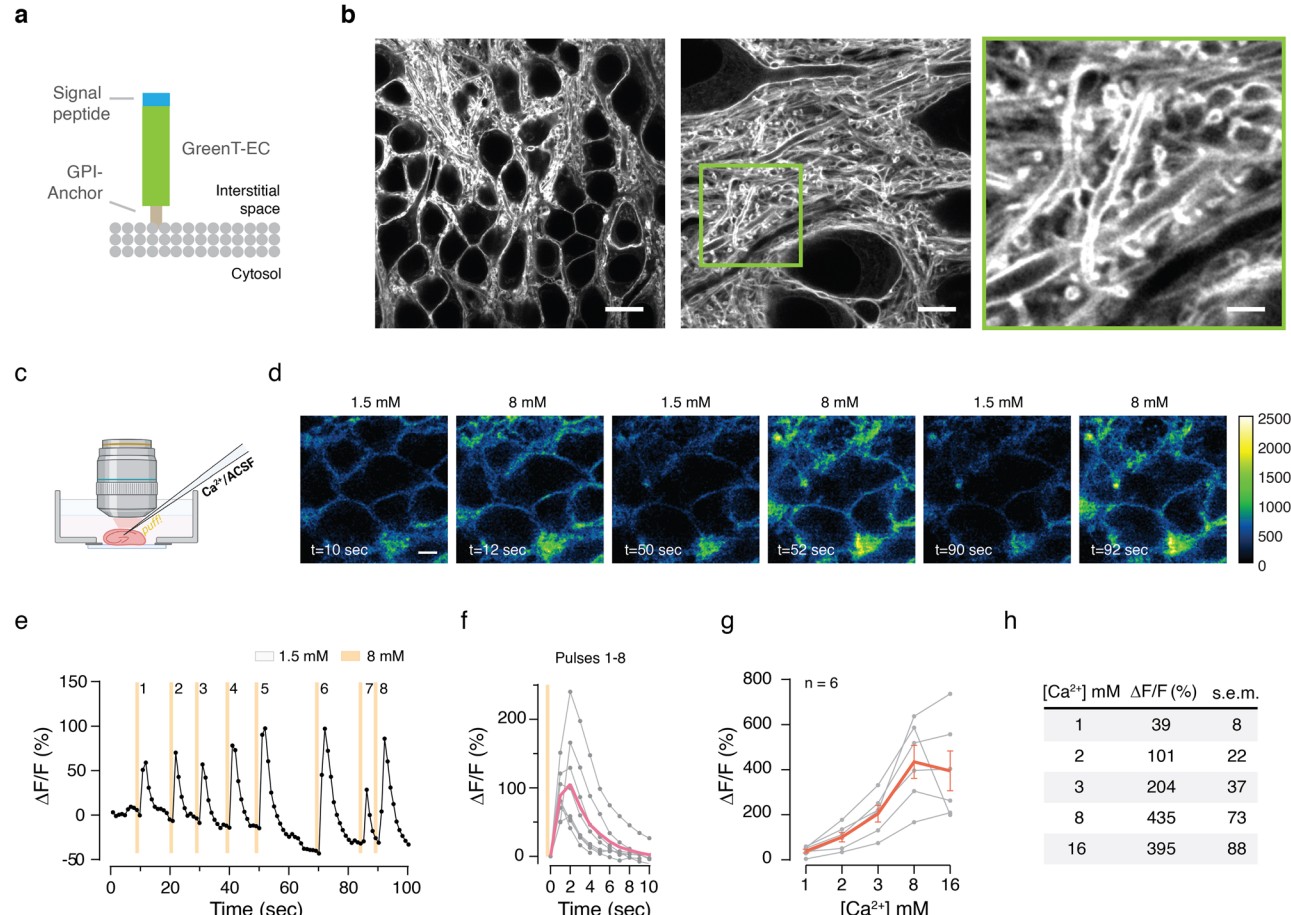

**Fig. 4 | Validation of GreenT-EC in rodent hippocampus. a** Scheme of surface targeted GreenT-EC (green) by means of an N-terminal signal peptide (blue) and a C-terminal GPI anchor (brown). **b** STED images of AAV-GreenT-EC infected neurons in hippocampal area CA1 of organotypic slice cultures (7 days postinfection). Scale bars: 10 µm, 5 µm, 2 µm, for left, middle and right panel, respectively. **c** Application of calcium solution to brain slices via micropipettes. **d** Time series representative images showing fluorescence changes in transfected hippocampal organotypic slices when exposed to 1.5 mM and 8 mM extracellular calcium. Scale bar 10 µm. **e** Dynamic GreenT-EC fluorescence changes upon repetitive brief (500 ms each) high-calcium (8 mM) solution puffs from baseline levels of 1.5 mM. Orange lines indicate the timing of the puffs. **f** Averaged response curves to the calcium injections from experiment **e**. **g** Calibration of GreenT-EC signals in hippocampal slices. The response was calculated using 0.5 mM $Ca^{2+}$ as the basal fluorescence value. Each gray line corresponds to a different slice ($n = 6$ biological replicates). The orange line represents the mean value at each calcium concentration (the error bar corresponds to the s.e.m.). **h** Summary table of mean responses and s.e.m. obtained in the calibration experiment presented in **g**. Source data are provided in the Source Data file.

(Supplementary Fig. 7c, d). This confirmed that the indicator affinity was well placed to report both potential decreases and increases in interstitial calcium. In addition, in organotypic brain slices kept in buffered medium at 1.5 mM $Ca^{2+}$, GreenT-EC signals dynamically increased when solutions containing 8 mM free calcium were locally puff-applied using a picospritzer (Fig. 4c–f, Supplementary Movie 1). Calibration experiments in organotypic brain slices revealed readily detectable changes in fluorescence over a physiological range of calcium concentrations (0.5 mM–3 mM; Fig. 4g), whereas the full dynamic range of GreenT-EC sensor in brain slices corresponds to a 4-fold change in relative fluorescence (Fig. 4g, h). We next evoked neuronal activity by electrically stimulating the Schaffer collaterals and recorded GreenT-EC responses in the CA1 projection area using two-photon microscopy (Fig. 5a, b). We detected spatially broad and seconds-scale increases in GreenT-EC signals after stimulation (Fig. 5b, Supplementary Fig. 8b), irrespective of imaging acquisition speed (Fig. 5b, Supplementary Fig. 8c). Similarly, spontaneous neuronal activity elicited increases in GreenT-EC signals exhibiting similar amplitudes, spread and kinetics (Fig. 5c and Supplementary Fig. 9). Next, we set out to explore the origins of these calcium rises in hippocampus using pharmacological manipulations. A combination of sodium orthovanadate (SOV, 5 µM) and benzamil hydrochlorate hydrate (BHH, 50 µM),

inhibitors of the plasma membrane calcium ATPase and the $Na^+/Ca^{2+}$ exchanger, essentially blocked the increases in interstitial calcium after Schaffer collateral stimulation (Fig. 5d), indicating that these signal rises are due to calcium extrusion into the ECS after neuronal activity. To further investigate the origin of the GreenT-EC rises, we treated the samples with tetrodotoxin (TTX, 1 µM) that blocks voltage-gated sodium channels, and thus prevents neurons from firing action potentials. Our results show that after TTX treatment, no GreenT-EC fluorescence increases upon stimulation can be detected (Fig. 5e). Similarly, treatment with $CdCl_2$ (100 µM), an inhibitor of voltage-gated calcium channels, shows the same results (Fig. 5f). Surprisingly, exposing the slices to cyanquixaline (CNQX, 20 µM), an antagonist of AMPA receptors has no effect on GreenT-EC signals upon stimulation (Fig. 5g). Taken together, our pharmacology experiments show that GreenT-EC fluorescence rises are elicited by neuronal activity (evoked or spontaneous) and most likely reflect calcium extrusion from neurons as part of a homeostatic response. Possibly, the compactness and size of the interstitial space could have an effect on shaping these signals.

**In vivo imaging of calcium homeostasis in transgenic zebrafish**
To establish a model of body fluid ionic homeostasis, we next generated transgenic zebrafish that stably expressed mCyRFP1-referenced

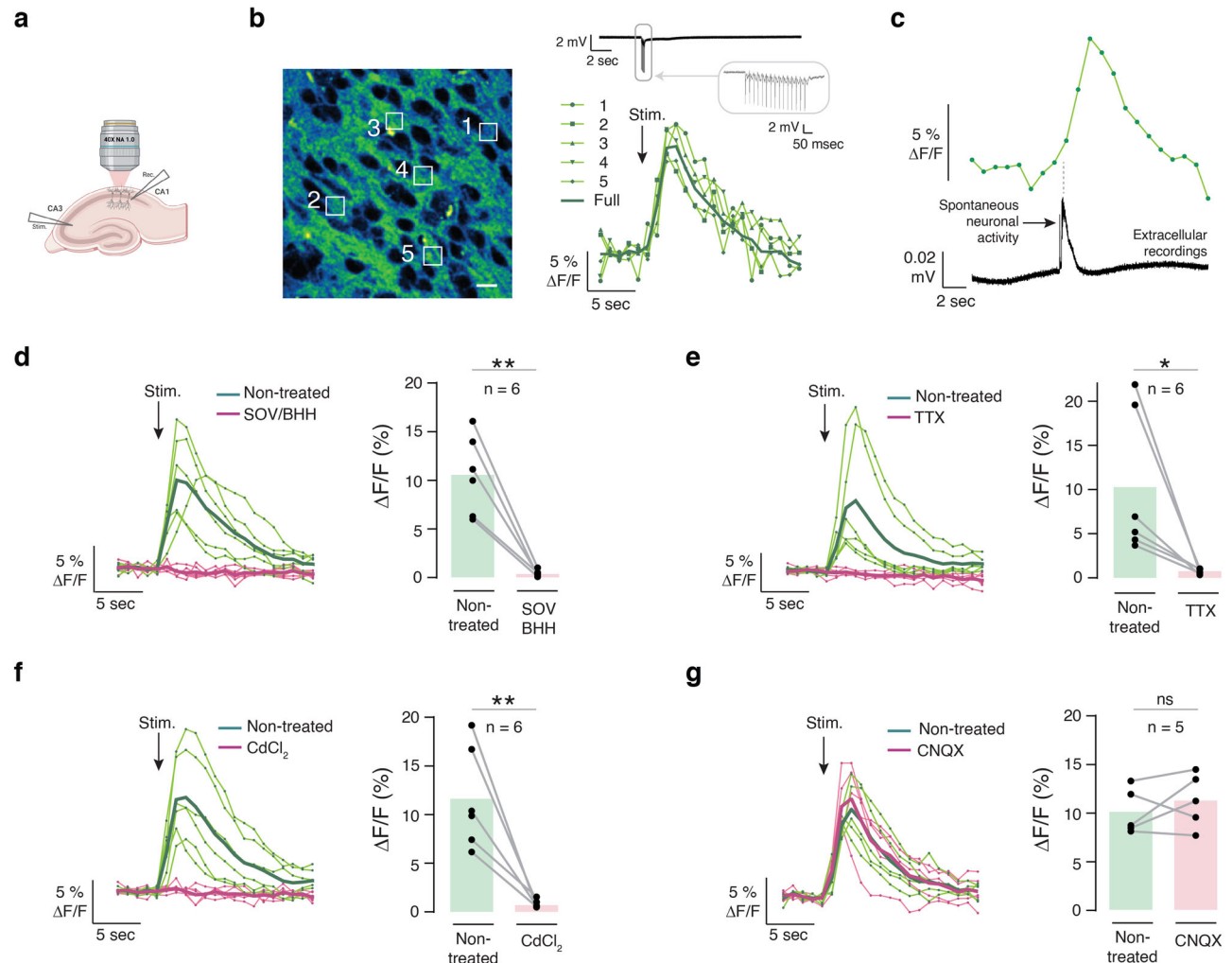

**Fig. 5 | Activity evoked hippocampal CA1 GreenT-EC signals. a** Electrical stimulation of the hippocampal Schaffer collateral pathway was used to elicit GreenT-EC responses monitored by two-photon microscopy. **b** Representative example of GreenT-EC expressing hippocampal organotypic slice (left) and the fluorescence traces upon electrical stimulation (right). White ROIs shown in the image correspond to individual traces. The dark-green trace (Full) correspond to a ROI covering the whole image. Scale bar 10 μm. **c** GreenT-EC fluorescence responses (top) upon spontaneous neuronal activity recorded extracellularly (bottom). **d** Activity evoked

hippocampal CA1 GreenT-EC signals and their inhibition by a combination of sodium ortho-vanadate (SOV, 5 μM) and benzamyl hydrochlorate hydrate (BHH, 50 μM) at 30 min after drug infusion ($n = 6$, $p = 0.0013$). **e–g** Activity evoked hippocampal CA1 GreenT-EC signals and their inhibition after 30 min application of TTX ($n = 6$, $p = 0.0362$), CdCl$_2$ ($n = 6$, $p = 0.0038$) or CNQX ($n = 6$, $p = 0.3976$). In all cases, n correspond to biological replicates and a parametric two-tailed paired t-test was used to evaluate the differences among control and treated groups (* $p < 0.05$, ** $p < 0.01$). Source data are provided in the Source Data file.

GreenT-EC semi-ubiquitously using the 10 kb *actb2* promoter[26] (Fig. 6). The transgenic fish displayed green and red fluorescence (Fig. 6a–c, e), outlining various cell types such as muscle cells, epithelial cells or vacuolar cells of the notochord in finest detail and without signs of aggregation of the indicator (Fig. 6c). For volumetric imaging the regulation of live tissue interstitial calcium, we performed confocal imaging of the dorsal fin fold and posterior notochord after various types of treatment (Fig. 6d). To validate the sensor functionality, we initially incubated 4-day post fertilization (dpf) zebrafish larvae acutely for 10 min in 3 mM EGTA. In vivo confocal imaging revealed a drastic loss of green fluorescence throughout tissue volumes, while red cytosolic mCyRFP1 fluorescence remained stable (Fig. 6e, f). Longer treatments with EGTA lead to considerable tissue disintegration. Time-lapse imaging of 4 dpf larvae treated with 5 mM EGTA revealed that loss of calcium in body fluids occurred within minutes (Supplementary Fig. 10). We next applied a model of hypocalcemia. In this, larvae are transferred to low-calcium water (0.03 mM), which is thought to induce a feedback loop that increases the capacity of the animal to absorb calcium via increased calcium-sensing receptor (CaSR)

expression and proliferation of CaSR-expressing cells[27,28]. We incubated transgenic larvae from 2 dpf in low calcium water enriched with the CaSR inhibitor Calhex231 (5 and 10 μM)[29] and imaged the fin fold at 4 dpf using in vivo confocal microscopy. Remarkably, treatment with Calhex231 led to a significant and dose-dependent decrease in interstitial calcium, although the effect was tissue-specific. We saw less pronounced effects in muscle tissue, possibly hinting at a less prominent role for CaSR in calcium homeostasis in muscle or a lower diffusion rate of interstitial calcium in muscle (Fig. 6e, g). Under similar experimental conditions and incubation periods Calcitriol (2.5 μM), the active form of Vitamin D and a hypercalcemic hormone in zebrafish[30], did not lead to detectable whole tissue changes of fluid calcium levels (Fig. 6h). Higher concentrations of Calcitriol led to animal death. Conversely, treatment with Dafadine A (12.5 μM), a CYP450 inhibitor that blocks the synthesis of the active form of Vitamin D[31], led to reduction in interstitial calcium levels (Fig. 6i, j). Finally, we were interested in exploring the natural capacity of larval zebrafish for maintaining calcium homeostasis under conditions of large variations in calcium concentrations in the external environment

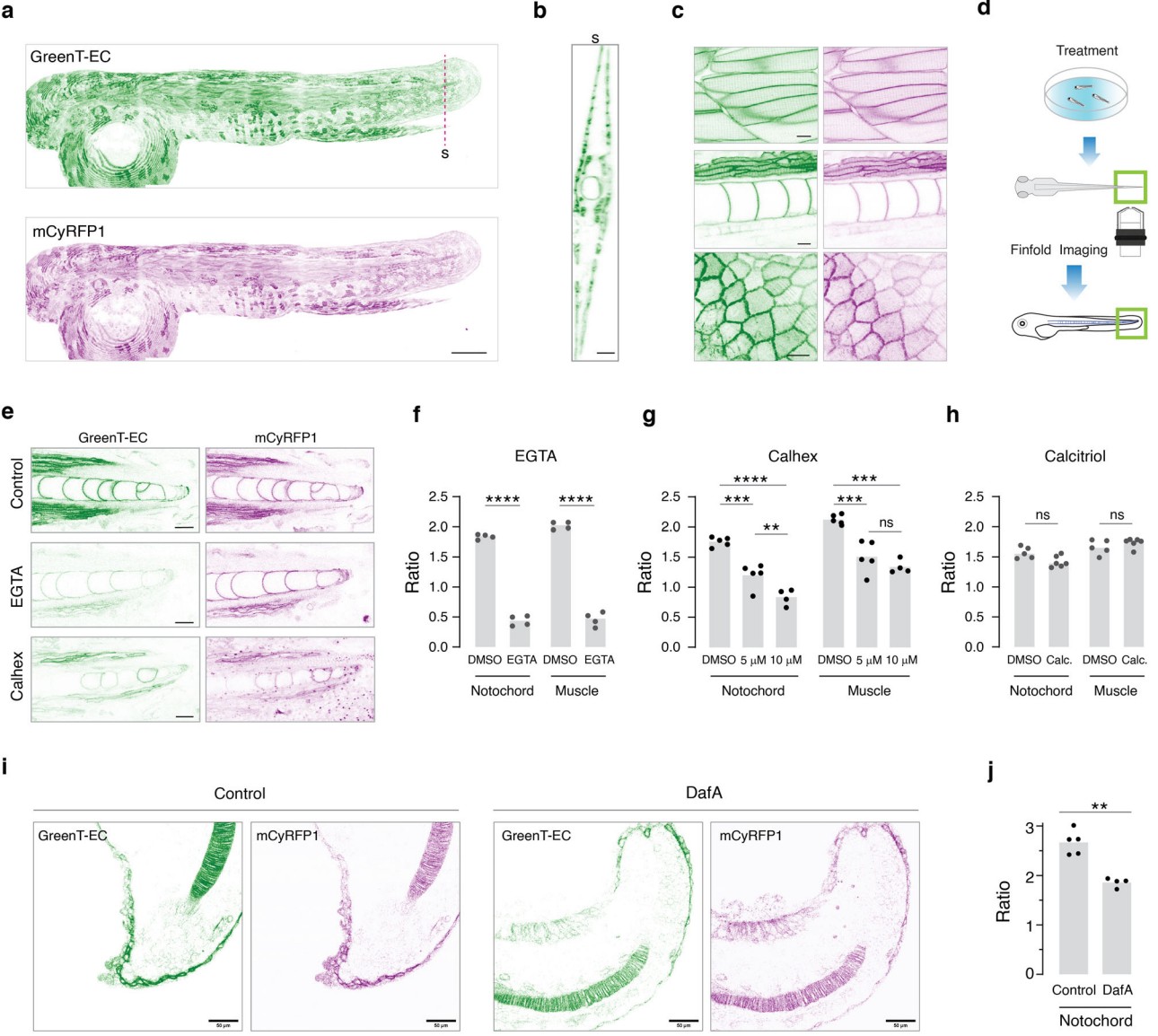

**Fig. 6 | In vivo imaging of calcium homeostasis in transgenic zebrafish.**
**a** Maximum intensity Z-projection of a confocal stack from a 2 dpf transgenic zebrafish expressing mCyRFP1-referenced GreenT-EC. Scale bar, 200 μm.
**b** Representative orthogonal view of the animal model in the fin fold. Scale bar, 20 μm. **c** Representative images of skeletal muscle cells (upper panel), vacuolar cells of the notochord and ventral/dorsal muscle (middle panel) and epithelial cells (bottom panel) in the fin fold. Scale bar, 10 μm. **d** Experimental strategy used for imaging transgenic GreenT-EC zebrafish larvae. After acute or chronic treatment fish are mounted in low melting agarose and the posterior fin fold is imaged. **e** Confocal images of the fin fold of 4 dpf zebrafish larvae after various treatments. Top: fish larvae under control conditions. Middle: after 10 min treatment with EGTA (3 mM). Bottom: treatment with Calhex231 (10 μM) from 2 dpf to 4 dpf. Scale bar, 20 μm. **f** Quantification of the EGTA effect ($n = 4$) on interstitial calcium as GreenT-EC/mCyRFP1 ratio around notochord and dorsal/ventral muscle cells.

**g** Effects of the CaSR inhibitor Calhex231 applied at 5 μM ($n = 5$) and 10 μM ($n = 4$) in E2 embryo medium containing 0.03 mM calcium for 48 h. Ratio values were compared with the control group ($n = 5$). **h** Effects of Calcitriol 2 μM ($n = 6$) treatment for 48 h on interstitial calcium around notochord and muscle compared to control animals ($n = 5$). **i** Confocal images of the fin fold of 1 dpf control zebrafish larvae (left) or treated with 12.5 μM DafadineA (DafA) (right) during 21 h. Scale bar, 50 μM. **j** Quantification of the DafA effect on interstitial calcium as GreenT-EC/mCyRFP1 ratio in the notochord ($n = 5$ for each group). In all experiments, n represents a biological replicate (fish). Two-tailed unpaired t-test were used to calculate the statistical significance between the control and treated groups (**f, h, j**). For multiple group analyses (**g**), One-way ANOVA followed by a Tukey multiple comparison test was performed (* $p < 0.05$, ** $p < 0.01$, *** $p < 0.001$ and **** $p < 0.0001$). Source data are provided in the Source Data file.

(Supplementary Fig. 11). We raised fish from birth in embryonic medium with large concentration differences in calcium (0.03 mM–10 mM, pH 7.4). When these fish were imaged at 2, 4 or 6 dpf no significant differences in body fluid calcium could be detected, demonstrating the remarkable ability of homeostatic mechanisms to clamp interstitial calcium levels to the physiological range. It is worth to highlight the stability of GreenT-EC/mCyRFP1 across animals considering that experiments were acquired over repeated sessions spread out over long periods of time.

## Discussion

GreenT-ECs are low-affinity calcium sensors with response characteristics tuned to match calcium concentrations present in extracellular body fluids. With the reported properties, they allow detecting relatively small calcium transients in a background of high residual concentrations in free calcium. GreenT-EC and its affinity appears optimal for the experiments we performed. However, GreenT-ECs may be brought to use in many different types of tissues and organisms. It cannot be ruled out that other calcium concentrations will be

encountered. We reasoned that experimenters might find it useful to have sensor alternatives with slightly higher and lower affinities available, should the need arise. These requirements are covered by GreenT-EC.b and GreenT-EC.c. The sensors were constructed by inserting a small fragment of a calcium-binding protein (68 amino acids) into a fluorescent protein. Thus, with appropriate level of engineering, large fractional fluorescence changes could be obtained without resorting to larger calcium-binding proteins, calcium-dependent protein interactions or the use of circularly permutated fluorescent protein backbones. As common with other fluorescent protein-based single fluorophore sensors, GreenT-ECs, with a pKa of 6.7 (Fig. 2), possess a residual pH-sensitivity in the physiological range. This should be kept in mind and controlled when larger pH changes are expected in a given experimental situation. The indicators were anchored on the surface of expressing cells, resulting in bright fluorescence and high signal-to-noise ratios for imaging. Sparse labeling or cell-type-specific labeling also will be possible this way. Surface labeling was implemented using two different strategies. In one example, we used the PDGF receptor transmembrane domain for anchoring and fused a C-terminal reference protein to the cytosolic intracellular side. This allows for convenient referencing of the green signal to the constant red cytosolic fluorescence. In a second example, we used GPI anchoring with the PPA signal peptide, which equally provided efficient surface localization (Fig. 3 and Supplementary Fig. 3). The absence of the reference protein here leaves spectral bandwidth for convenient combination with other types of fluorophores to label subcellular structures, for use with red fluorescent cytosolic calcium probes or with other types of red sensors, etc.

As alternative to surface localization, cells could be engineered to secrete GreenT-ECs into the extracellular fluid. This might, however, require substantial ubiquitous expression, as sensors might become diluted within body fluids. Engineered binding to components of the extracellular matrix might be favorably in this case. Fusions to collagen-binding domains were shown to work, in principle, with FRET-based "Twitch" calcium biosensors[32]. A secondary application for GreenT-ECs might be its use as a tag to image surface exposure of proteins or secretion, given the bright fluorescence when exposed to high calcium extracellular fluids and dimness when residing in the cytosol (Fig. 3, Supplementary Fig. 3, 4).

We validated GreenT-ECs in primary hippocampal neurons. Both in dissociated neurons and in organotypic slices we could show bi-directional fluorescence changes that faithfully reproduced the extracellular decreases or increases in free calcium. We found activity-induced (both evoked and spontaneous) increases in interstitial calcium that appeared driven by cellular extrusion mechanisms, by neurons and possibly by astrocytes as well[33]. Transient decreases in interstitial calcium due to the influx of calcium into the large cytosolic volumes upon neuronal activation were not detected, which may be highly local and too rapid for our imaging system. GreenT-ECs with their fast off-rate should otherwise be well suited to report such decreases. Conceivably, the specific distribution of channels, pumps and the dimensions of the intercellular space could contribute to shaping the time course and spatial spread of such signals. Dynamics in brain extracellular calcium had been described before[34,35]. Calcium extrusion from cells was documented using ion selective electrodes in epithelial tissues[36]. Use of membrane-anchored synthetic calcium dyes similarly revealed calcium extrusion in cultured cells in vitro[37,38].

Transgenic zebrafish lines stably expressing GreenT-EC appear as a promising model of body fluid calcium homeostasis. Using in vivo confocal imaging, we could show the effects of inhibitors of the CaSR and of CYP450 on uncoupling feed-back loops necessary for maintaining physiological calcium concentrations in body fluids. Similar studies may contribute to elucidate mechanisms and players in the regulation of physiological levels of free calcium in body fluids and

their potential compartmentalization across tissues. Dis-regulation of interstitial free calcium is linked to numerous medical disorders in humans[39,40]. While further investigations into these aspects were outside the scope of this study, these data show the potential of GreenT-ECs for new discoveries in physiology.

## Methods

### Molecular biology
The plasmid vector pRSET-B (Invitrogen) was used for the library generation, bacterial screenings and protein production. pcDNA 3.1+ (Invitrogen) and pCAGIG (Addgene, USA) were the expression vectors used for the studies in mammalian cells. All constructs were cloned using the homology-based SLiCE methodology[41]. Briefly, fragments were amplified by PCR containing 20 nucleotides overhangs overlapping with the required upstream and downstream sequences. SLiCE extract was prepared as suggested by the authors and 10 µl reactions were used with a vector/insert ratio of 1:1. E. coli XL1 Blue cells (Invitrogen) were used for all cloning steps and library screenings. Error-prone PCR mutagenesis was performed using the GeneMorph II Random mutagenesis kit (Agilent, Germany). Standard PCR reactions for cloning or site-mutagenesis were done using Herculase II Fusion polymerase (Agilent, Germany). All primers were ordered from Eurofins Genomics, Germany.

### Directed evolution
The DNA libraries were incorporated into E. coli XL1 Blue cells and transferred onto LB-Agar plates supplemented with ampicillin (Roth), obtaining 500–800 colonies per plate. Depending on the size of the library, 20–50 plates were screened in each round. After an overnight incubation at 37 °C, LB-agar plates containing the libraries were further incubated at 4 °C during 24 h prior imaging to favor protein folding. Using a home-built screening platform[42], the plates were imaged before and after the addition of 10 ml of screening buffer (30 mM MOPS, 100 mM KCl, 50 mM $CaCl_2$, 0.5%, agar pH 7.2) maintained at 42 °C during the screening. Images were acquired and analyzed using a customized python routine[42]. In general, after each round of screening, the selected indicators were expressed and purified for further characterizations. The clones were filtered first according to its response to calcium, then its extinction coefficient, and finally if a suitable candidate was identified, its response was evaluated in mammalian cells. Calibrations and comparisons were made using GCaMP6s as a control indicator. Once an acceptable response to calcium was obtained (>3000 %), new rounds of screenings were done focusing on improving protein expression, adjusting the kinetic performance and the calcium-binding affinity. Further details on the strategy for directed evolution of GreenT-ECs are given in Supplementary Note.

### Protein purification
Proteins with histidine-tags were expressed in E. coli BL21 (Invitrogen) overnight at 37 °C in 50 ml auto-inductive LB (LB supplemented with 0.05% D-(+)- glucose (w/v), 0.2% lactose (w/v), 0.6% glycerol (v/v)). Bacteria were harvested by centrifugation (4 °C, 10 min, 6000 x g) and re-suspended in 10 ml Resuspension buffer (20 mM $Na_2PO_4$, 300 mM NaCl, 20 mM imidazole) (Sigma Aldrich) supplemented with protease inhibitors (4 µM PMSF, 20 µg/ml Pepstatin A, 4 µg/ml Leupeptin) (Sigma Aldrich), 5 µg/mL DNase and 10 µg/ml RNAse (Sigma Aldrich). Bacteria were first lysed physically through sonication on ice for 7 min 0.8 cycle, 0.8 power output (Bandelin Sonoplus). Insoluble components were pelleted through centrifugation (4 °C, 30 min at 20,000x $g$). For purification, the supernatant was incubated with 150 µl 6% (v/v) Nickel-IDA agarose bead suspension (Jena Bioscience) for 2 h at 4 °C under mild agitation. Agarose beads were collected in 1 ml propylene gravity flow columns (Qiagen) and washed with 10 ml Resuspension buffer. The proteins were collected using 700 µl elution buffer (20 mM $Na_2PO_4$, 300 mM NaCl, 300 mM

imidazole) (Sigma Aldrich) and dialyzed against MOPS buffer (30 mM MOPS, 100 mM KCl, pH 7.2).

## In vitro spectroscopy

The fluorescence change ($\Delta F/F_0$) of the indicators was determined in 96-well plates using a fluorescence plate reader (Tecan). The $Ca^{2+}$ free fluorescence ($F_0$) was measured in MOPS buffer supplemented with EGTA 0.4 mM. The fluorescence of the indicator corresponding to the $Ca^{2+}$ bound state was measured in MOPS buffer supplemented with 100 mM $Ca^{2+}$ and 1 mM $Mg^{2+}$.

The molar extinction coefficient (EC) was determined through the absorption of the denatured chromophore at 452 nm (extinction coefficient 44 mM$^{-1}$cm$^{-1}$). Proteins were prepared in MOPS buffer supplemented with 60 mM $CaCl_2$ and the absorbance spectrum was acquired before and after the addition of NaOH to a final concentration of 0.025 M.

The quantum yield of new variants was determined relative to mNeonGreen by using the slope method. First, the absorption and emission spectra of three serial 1:2 dilutions were acquired in the same cuvette. Then, the integrated emission spectrum was plotted against the maximum absorption and the slope was determined. For mNeon-Green a quantum yield of 0.8 was adopted[20]. The $Ca^{2+}$ affinity of the indicators was determined using MOPS buffer supplemented with 10 mM EGTA, 1 mM $Mg^{2+}$ and increasing concentrations of $Ca^{2+}$ as previously described[43]. Dissociation constant (Kd) values were determined by plotting the log10 values of the [$Ca^{2+}$] free concentrations in mol/l against the corresponding $\Delta F/F_0$ values (normalized to $\Delta F/F_0$ at 39.8 μM $Ca^{2+}$), and fitting a sigmoidal curve to the plot. Prism was used for data analysis.

To determine the pKa of a fluorescent protein (indicating its pH stability), a series of MOPS/MES buffered solutions supplemented with 100 mM $Ca^{2+}$ were prepared, with pH values adjusted in 0.5 pH steps from pH 5.5 to pH 8.5 using NaOH and HCl. In a bottom 96 well plate, triplicates of 200 μl of buffer containing 0.5–1 μM of protein were prepared for each pH value and incubated for 10 min at room temperature. Subsequently, all emission spectra were recorded. To determine the pKa value, the relative fluorescence values at the protein's emission maximum were plotted against the pH values and a sigmoidal fit was applied.

To determine the kinetic rates of the calcium indicators, a Cary Eclipse fluorescence spectrophotometer (Varian) fitted with an RX pneumatic drive unit (Applied Photophysics) was used. For obtaining the macroscopic off-rate constant (Koff), two stock solutions were prepared as follows: a calcium-saturated indicator solution (30 mM MOPS, 50 mM $CaCl_2$, 2 mM $MgCl_2$, 100 mM KCl, ~0.2–1 μM indicator, pH 7.2) and a BAPTA solution (30 mM MOPS, 100 mM KCl, 100 mM BAPTA, pH 7.2). The stopped-flow experiment was carried out at room temperature (~23 °C) and the two solutions were mixed with an injection pressure of 3.5 bar. Excitation was set to 480 nm and emission was detected at 520 nm. The acquisition time was set to 12.5 ms, duration to >10 s and mixing volume to 400 μl with a mixing dead time of the instrument of 8 ms. The decay time (τ, s) was determined by fitting with a double-exponential curve to the fluorescence response using Prism. Macroscopic on-rate kinetics (K$_{obs}$) were obtained by mixing the calcium-free buffer containing the protein (30 mM MOPS, 100 mM KCl, 1 mM $MgCl_2$, ~0.2–1 μM indicator, pH 7.2) and solutions containing increasing concentrations of $CaCl_2$ (30 mM MOPS, 100 mM KCl, 0.1–100 mM $CaCl_2$, 1 mM $MgCl_2$, ~0.2–1 μM indicator, pH 7.2). Concentrations of free calcium were calculated using WEBMAXC STANDARD server.

## Crystallization and structure determination

Crystals of the GreenT-EC intermediate variant named NRS 1.2 were formed using the sitting drop method. The precipitant solution was 0.2 M ammonium sulfate with 30% w/v PEG 400. Droplet conditions were: 0.4 μL total volume, 200 nL protein + 200 nL precipitant. The crystals were cryoprotected with addition of 30% of ethylene glycol and flash-frozen in liquid nitrogen. X-ray data sets were recorded on the 10SA (PX II) beamline at the Paul Scherrer Institute (Villigen, Switzerland) at wavelength of 1.0 Å using a Dectris Eiger 16 M detector with the crystals maintained at 100 K by a cryocooler. Diffraction data were integrated using XDS (BUILT = 20220220) and scaled and merged using AIMLESS (0.7.7); data collection statistics are summarized in Supplementary Table 1. Initially the NRS 1.2 data set was automatically processed at the beamline to 1.3 Å resolution and a structure solution was automatically obtained by molecular replacement using pdb 5MWC as template. The map was of sufficient quality to enable 90 % of the residues expected for NRS 1.2 to be automatically fitted using Phenix autobuild. The model was finalized by manual rebuilding in COOT2 (0.9.6) and refined using in Phenix refine (1.19.2).

## Cell lines and tissue culture

HeLa and HEK 293 T (Invitrogen) cells were grown in high glucose Dulbecco's Modified Eagle Medium with high glucose, pyruvate (Gibco) supplemented with 10% fetal bovine serum (FBS, Gibco), 29.2 mg/ml of L-glutamine, 10,000 units of penicillin and 10,000 μg streptomycin at 37 °C with 5% $CO_2$.

## Evaluation of indicator surface display

Different combinations of signal peptides and membrane domains were evaluated in HeLa cells. The cells were seeded in 35 mm glass-bottom dishes (Mat-Tek) pre-coated with Poly-L-lysine (Sigma) and transfected with the different constructs one day before imaging, according to the manufacturer instructions (Lipofectamine 3000). Hoechst staining was performed immediately before imaging using a final concentration of 2 μg/mL for 20 min. The imaging buffer used for these experiments was Hank Balanced Salt Solution (HBSS) supplemented with 1 mM $MgCl_2$ and 3 mM $CaCl_2$. Images were acquired in a Leica Stellaris5 microscope with a 60x oil-immersed objective. Different export signal peptides and transmembrane domain were evaluated: IβCE (from Integrin-β of C. elegans)[44,45], Igk (from human CH29 light chain)[44,46], Nrxn1 (mouse Neurexin I first 63 aa)[44,46], PPA (N-terminal 24 aa of mouse preproacrosin signal peptide)[25], PDGFRB (Homo sapiens platelet derived growth factor receptor beta)[47], GPI (mouse Thy-1 glycosylphosphatidylinositol) anchoring domain[25], NXN (rat neurexin-1β) and NLG (rat neuroligin-1)[44,46].

## Evaluation of reference proteins using Igk-PDGFR surface localized GreenT-EC

HeLa cells were seeded in 35 mm glass-bottom dishes (Mat-Tek) and transfected with the different constructs according to the manufacturer instructions (Lipofectamine 3000). The imaging buffer used for these experiments was Hank Balanced Salt Solution (HBSS) supplemented with 1 mM $MgCl_2$ and 3 mM $CaCl_2$. Images were acquired in a Nikon Spinning Disk confocal microscope with a 20x oil-immersed objective. In each case, the reference proteins mCarmine[42], mScarlet[48], mCyRFP1[24], mCerulean3[49] and mTurquoise2[50] were inserted in the C-terminal extreme of the PDGFR domain, separated by a short linker. The mCyRFP1 referenced construct was compared to a cytosolic version (not bearing any export signal or transmembrane domain). In this case, mCyRFP1 was cloned at the N-terminus of GreenT-EC or GCaMP6f using a 20 aa hydrophobic flexible linker as previously described[51]. Cells were permeabilized using ionomycin (2.5 μM), which allowed the entry of extracellular calcium (1.5 mM), or stimulated with Histamine (200 μM) to induce a physiological increase in cytosolic calcium.

## Affinity titrations on the surface of HEK 293 T cells

HEK293T cells were seeded on day one into 35 mm glass-bottom dishes (Mat-Tek) pre-coated with Poly-L-lysine (Sigma). On day three, the cells were transfected using 1 μg of the plasmids coding for the

different variants according to the manufacturer instructions (Lipofectamine 3000). For each variant, three dishes were prepared. On day four, cells were washed with buffer MOPS 30 mM KCl 100 mM, EGTA 0.4 mM, pH 7.2 during 2 min. Then 3 ml of buffer MOPS 30 mM KCl 100 mM, pH 7.2 was added and cells were imaged in a Leica SP8 confocal microscope using a 63x water-immersed objective. For the titration, 80–100 μl of $CaCl_2$ stock solutions were added drop-wise to the dishes during time-lapsed imaging. Normally, images were acquired every 1 min.

## Flow cytometry

HeLa cells were seeded in 35 mm dishes (Corning) and transfected with the different constructs according to the manufacturer instructions (Lipofectamine 3000). Twenty-four hours after transfection, cells were detached with Versene solution (Gibco, ThermoFisher) and collected in HBSS supplemented with $CaCl_2$ 3 mM to inactivate the detachment agent. After centrifugation at 1000 g during 5 min, cells were resuspended in 1.5 ml MOPS buffer supplemented with 3 mM $CaCl_2$ / 1 mM $MgCl_2$, and transferred to 5 ml round bottom tubes with cell strainer snap cap (Corning). All samples were evaluated in an Attune NxT Analyzer cytometer (ThermoFisher Scientific) before and after the addition of EGTA 5 mM. At least 100.000 events were recorded for each condition and further analyzed using FlowJo v10.8.1 (FlowJo, LLC). Events were first gated by forward and side scattering for selecting cells and excluding doublets as shown in the figures. Attune BL1-A channel (Excitation 488 nm - Emission 530/30 nm) was used for monitoring GreenT-EC, VL1-A (Excitation 405 nm - Emission 440/50 nm) for mTurquoise2 and mCerulean3 constructs, YL2-A (Excitation 561 nm - Emission 620/15 nm) for mScarlet, and YL3-A (Excitation 561 nm - Emission 695/40 nm) for mCarmine. Mean fluorescence values were used to calculate the response as: $(F_{Ca2+}-F_{EGTA})/F_{EGTA}$. Data was plotted using GraphPad Prism and both ungated and gated full populations are submitted separately in an excel file.

## Image processing and ratioing

The general processing procedure for mammalian cell titrations and zebrafish experiments consisted in: i) Noise reduction (typically using a median or Gaussian filter of 2 pixels), ii) Thresholding saturated pixels and noise/cytoplasm signals. During this process, the original image was divided by the Threshold mask to assign N/A values to the pixels that were thresholded out. This avoids problems arising from averaging cero-value pixels during the ROI analysis. iii) Finally, the processed GreenT-EC and mCyRFP1 channels were used to obtain a ratiometric image GreenT-EC/mCyRFP1. This method allowed us to monitor the ratio signals in the membrane with minor interference from background or low-intensity cytosolic signals. ImageJ macro routines were generated to perform the image processing, ROI detection/drawing, measurement of individual channels and ratio, and exporting of the results. The data analysis was done in GraphPad. Images were displayed with white background in cases where high contrast was required to clearly visualize the different structures[52]. The ImageJ scripts used can be found in the Source Data file.

For brain slices and dissociated neuronal culture experiments, all images were processed and analyzed using ImageJ software (NIH). For time-lapse 2 P acquisitions, bleach correction based on an exponential fit provided by ImageJ was performed. Since a reference fluorescent protein was not included in these experiments, no rationing was performed.

## Primary neuronal cell culture

All experiments were performed in accordance with the European directive on the protection of animals used for scientific purposes (2010/63/EU). Banker culture of hippocampal neurons were prepared from 18 day pregnant embryonic Sprague-Dawley rats as described

previously[53]. Briefly, hippocampi were dissected in HBSS containing Penicillin-Streptomycin (PS) and HEPES and dissociated with Trypsin-EDTA/PS/HEPES. Neurons were plated in minimum essential medium supplemented with 10% horse serum on coverslips coated with 1 mg/ml poly-L-lysine in 60 mm petri dishes at a density of 250000 cells per dish. Following neuronal attachment, the coverslips were flipped onto 60-mm dishes containing a glial cell layer in Neurobasal-Plus medium supplemented with GlutaMAX (GIBCO, #35050-038) and B27-Plus Neuronal Supplement (GIBCO, A3653401). Cells were maintained at 37 °C with 5% $CO_2$ for 13–15 days. Neurons were transfected at DIV 7–9 using the calcium-phosphate co-precipitation method. Per dish, precipitates containing 10 μg plasmid DNA of GreenT-EC were prepared using the following solutions: TE (1 mM Tris–HCl pH 7.3, 250 mM EDTA pH 8), $CaCl_2$ (2.5 mM $CaCl_2$ in 10 mM HEPES, pH 7.2), 2 x HEPES-buffered saline (HEBS; 11 mM D-Glucose, 42 mM HEPES, 10 mM KCl, 270 mM NaCl and 1.4 mM $Na_2HPO_4.2H_2O$, pH 7.2). Coverslips containing neurons were moved to 12 well multi-well plates containing 450 μl/well of conditioned culture medium. The 50 μl precipitate solution was added to each well, in the presence of 2 mM kynurenic acid (Sigma-Aldrich #K3375) and incubated for 1h-1h30 at 37 °C. Then, cells were washed with nonsupplemented Neurobasal medium containing 2 mM kynurenic acid for 20 min and moved back to their original culture dish. Cells were imaged at DIV 13–15.

## Organotypic hippocampal slice culture

Mice were housed under a 12 h light/12 h dark cycle at 20–22 °C with ad libitum access to food and water in the animal facility of the Interdisciplinary Institute for Neuroscience (University of Bordeaux/CNRS), and monitored daily by trained staff. All animals used were free of any disease or infection at the time of experiments. Pregnant females and females with litters were kept in cages with one male. We did not distinguish between males and females among the perinatal pups used for organotypic cultures, as potential anatomical and/or physiological differences between the two sexes were considered irrelevant in the context of this study. C57Bl/6 J wild-type mice were used for all brain slice experiments in this study. Brain organotypic cultures were prepared according to the method described previously[54]. Briefly, hippocampal slices were obtained from postnatal P5-7 old C57Bl/6 J mouse pups. The animals were quickly decapitated and hippocampi were dissected out in cold GBSS (ThermoScientific) with 10 mM Glucose (VWR) and 2 mM kynurenic acid (Sigma-Aldrich). They were sliced on a McIlwain tissue chopper to generate coronal brain slices of 350 μm thickness. After 30 min of incubation at 4 °C, the slices were transferred on sterilized hydrophilic polytetrafluoroethylene (PTFE) membrane (FHLC04700; Sigma-Aldrich, 0.45 μm pore size) pieces, which were placed on top of cell culture inserts (PICMORG50; Sigma-Aldrich 0.4 μm pore size). The inserts were held in a 6-well plate filled with 1 ml of medium (50% Basal medium eagle (BME), 25% Hank's Balanced Salt Solution (HBSS, pH 7.2), 25% Horse Serum, 11.2 mmol/L glucose and 20 mM glutamine; all from Thermofisher) and cultured up to 14 days at 35 °C/5% $CO_2$. Culture medium was replaced every two days.

## AAV production and injections

In order to introduce the GreenT-EC plasmid to neurons in organotypic brain slices, the GreenT-EC coding sequence was first inserted into a specific viral plasmid (Addgene #50477) with a neuron-specific promotor CaMKII. After testing the plasmid for its correct size and integrity via restriction enzyme digestion, the construct was sent to in-house virus production facility, which provided us with a construct of GreenT-EC integrated into AAV9 particles. The viral particles were injected into the hippocampal slices via microinjections using a glass pipette connected to Picospritzer (Parker Hannifin). Briefly, the virus was injected via a pipette positioned into the CA1 area of the slice by brief pressure pulses (30 ms; 15 psi). The virus was injected at least 7 days prior to the experiments.

## STED and confocal microscopy of hippocampal neurons

We used a custom-built STED/confocal setup[55] constructed around an inverted microscope body (DMI 6000 CS, Leica Microsystems) which was equipped with a TIRF oil objective (x100, 1.47 NA, HXC APO, Leica Microsystems) and a heating box (Cube and Box, Life Imaging Services) to maintain a stable temperature of 32 °C. A pulsed-laser (PDL 800-D, PicoQuant) was used to deliver excitation pulses at 488 nm and a de-excitation laser (Onefive Katana 06 HP, NKT Photonics) operating at 594 nm was used to generate the STED light pulses. The STED beam was profiled to a donut shape using a spatial light modulator (Easy3D Module, Abberior Instruments). Image acquisition was controlled by the Inspector software (Abberior Instruments). The spatial resolution of the microscope was 175 nm (x-y) and 450 nm (z) in confocal mode and 60 nm (x-y) and 160 nm (z) in STED mode.

## Confocal/-STED imaging of hippocampal neurons in dissociated cultures and organotypic brain slices

For imaging, either slices or dissociated neurons were transferred on their glass coverslip to an imaging chamber and immersed in an imaging medium (in case of slices: artificial cerebrospinal fluid, ACSF; consisted of (in mM) 119 NaCl, 2.5 KCl, 1.3 $MgSO_4$, 1 $NaH_2PO_4$ x 2H2O, 1.5 $CaCl_2$* x 2H2O, 20 D-Glucose x $H_2O$ and 10 HEPES (all from Sigma Aldrich); 280 mOsm; pH 7.4; in case of dissociated neurons: Tyrode solution; consisted of (in mM) 10 D-Glucose x $H_2O$, 100 NaCl, 5 KCl, 2 $MgCl_2$ x 6H2O, 25 HEPES, 2 $CaCl_2$* x 2H2O). Confocal time-lapse images of dissociated neurons had a field of view: 50 μm x 50 μm. STED single-plane images of organotypic brain slices were either 100 μm x 100 μm or 50 μm x 50 μm with a pixel size of 48,6 nm, 0.3 ms dwell time. The excitation power was 0.5 μW (measured before the objective) and the STED power was 30 mW. *The concentration of calcium in the solutions varied depending on the experiment.

## Two-photon microscopy

We used a commercial two-photon microscope (Prairie Technologies) and 40X water immersion objective (NA 1.0; Plan-Apochromat, Zeiss). For GreenT-EC excitation, a two-photon laser (Ti:sapphire, Chameleon Ultra II; Coherent) was tuned to 850 nm; with laser power ranging from 10–25 mW in the focal plane. The fluorescence signal was collected in a nondescanned manner by PMT detectors. The imaging parameters were adjusted using the commercial software provided by Prairie (Prairie View). For image acquisition, individual slices were transferred to a submerged recording chamber filled with HEPES-based ACSF solution (in mM): 119 NaCl, 2.5 KCl, 1.3 $MgSO_4$, 1 $NaH_2PO_4$ x 2H2O, 20 D-Glucose x $H_2O$, 10 HEPES and varying $CaCl_2$ concentrations (all from Sigma Aldrich); 300 mOsm; pH 7.4. For most of the experiments, we acquired time-lapse images of single planes (126 μm x 126 μm) every 1 s for a total number of 40 repetitions. If the imaging parameters differed, it is mentioned in the respective figure legend.

## Calcium puffing experiments

In order to locally apply calcium solutions, we inserted a glass micropipette into the slice expressing GreenT-EC sensor. By injecting brief high-pressure pulses (10–100 ms, 15 psi) via a Picospritzer (Parker Hannifin), ACSF solutions of varying calcium concentrations were locally delivered into the region of interest. To avoid pressure-induced z-drift, the injection parameters, pulse duration and pressure level, were optimized.

## Electrophysiology

Schaffer collateral fibers in hippocampal slices were electrically stimulated and evoked field excitatory postsynaptic potentials (fEPSP) were recorded in the *stratum radiatum* of hippocampal CA1. Two glass micro-electrodes (tip resistance 5-6 MΩ) for stimulation and recording were filled with ACSF and carefully positioned in the slice and placed at depths where imaging was performed. The current pulses (5–15 pulses,

0.2-0.3 ms in duration) were delivered via the stimulating electrode from a stimulus isolator (AMPI; Science Products). The stimulus strength varied between 10–40 μA. Field potentials were recorded using a patch clamp amplifier (Multiclamp 700B; Molecular Devices).

To block calcium extrusion from cells, we applied two calcium-pump blockers: 5 μM sodium-orthovanadate (SOV, Sigma Aldrich) and 50 μM benzamil hydrochloride hydrate (BHH, Sigma Aldrich) dissolved in HEPES-based ACSF. To block voltage-gated sodium channels, we applied 1 μM tetrodotoxin (TTX, Hello Bio). Inhibition of voltage-gated calcium channels was done using 100 μM $CdCl_2$ (Sigma Aldrich). Finally, 20 μM cyanquixaline (CNQX, Sigma Aldrich) was used to block AMPA receptors.

## Zebrafish strains and husbandry

Transgenic zebrafish derived from the outbred AB strain were used in all experiments. Zebrafish were raised under standard conditions at 28 °C. Animals were chosen at random for all experiments. Zebrafish husbandry and experiments with all transgenic lines were performed under standard conditions as per the Federation of European Laboratory Animal Science Associations (FELASA) guidelines[56], and in accordance with institutional (Université Libre de Bruxelles (ULB)) and national ethical and animal welfare guidelines and regulation, which were approved by the ethical committee for animal welfare (CEBEA) from the Université Libre de Bruxelles (protocols 578N-579N). To generate the actb2:GreenT-EC construct, the vector containing 9.8 kb of zebrafish b-actin2 (*actb2*) promoter[26] was digested by SpeI/NotI and GreenT-EC cloned using the previously mentioned SLiCE methodology[41]. Transgenics were generated using the I-SceI meganuclease system. Two founders were isolated and screened for the strength of green and red fluorescence in the body. The line with brighter expression was used to perform all experiments, and was designated *Tg(actb2:GreenT-EC)*[ulb18].

For experiments, transgenic zebrafish larvae, *Tg(actb2:GreenT-EC)*, were obtained from outcross of the transgenic adult animal to AB wild-type animals. Larvae between 2 and 6 dpf (days post-fertilization) were used in all experiments. Zebrafish have indeterminate growth with the rate varying with parameters such as fish density, so larvae were kept at a density of 1 larvae per 1 ml of E2 medium (7.5 mM NaCl, 0.25 mM KCl, 0.5 mM $MgSO_4$, 75 mM $KH_2PO_4$, 25 mM $Na_2HPO_4$, 0.35 mM $NaHCO_3$) supplemented with either 0.3, 0.03, 2 or 10 mM $CaCl_2$, 0.5 mg/L methylene blue, pH 7.4 since birth unless specified otherwise. Anesthesia was administered in E2 medium using 0.02% pH 7.0 tricaine methanesulfonate (MS-222; E10521; Sigma-Aldrich, Darmstadt, Germany).

## Zebrafish confocal image acquisition

Animals were anesthetized in 0.02 % tricaine methanesulfonate (MS-222; E10521; Sigma-Aldrich, Darmstadt, Germany) and mounted in 1% Low-Melt Agarose (50080; Lonza, Basel, Switzerland) and imaged on a glass-bottomed FluoroDish (FD3510–100; World Precision Instruments (WPI), Sarasota, Florida) using a LSM 780 confocal microscope (Zeiss). Finfold and muscle were imaged using a 40x/1.1 N.A. water correction lens. Imaging frame was set at 1024 × 1024, and the distance between confocal planes was set up at 3 μm for Z-stack cover, on average, a thickness of 60 μm. Samples were excited with 488 nm laser and fluorescence was collected in the two channels simultaneously using detector width of 493–530 nm for GreenT-EC and 550–740 nm for mCyRFP1.

## Zebrafish pharmacological treatments

All compounds for treating embryos were dissolved in DMSO according to manufacturer, diluted in E2 embryo medium (0.3 mM or 0.03 mM $Ca^{2+}$), changed daily and embryos treated by immersion. The compounds, and concentrations used, with catalogue numbers were: EGTA (Roth), 0.3 mM for 10 min; Calhex 231 hydrochloride (SML0668-

Sigma), 5 or 10 μM for 48 h (from 2 to 4 dpf); Calcitriol (C3078-TCI) 2.5 μM for 24 h (3 to 4dpf). Dafadine A (HY-16670-MedChemExpress) was applied at 12.5 μM from 3 to 24 h postfertilization.

## Time-lapse imaging of zebrafish

For EGTA time-lapse imaging, 4 dpf larvae were immersed in 5 mM EGTA solution, followed by embedding in 1% low-melt agarose. Subsequently, the solidified agarose was covered with 1 ml of 5 mM EGTA solution.

## Statistics and reproducibility

Representative pictures were included only after systematic repetitions along different experimental replicates, and this can be evidenced from the control groups, acquired multiple times and for every drug treatment or condition, both for zebrafish and brain slices experiments. In the case of mammalian cell images, they were supported by biological replicates and additional evidences using flow cytometry. In the case of purified proteins, during the development of the project we observed that the measurements of the biophysical parameters were highly reproducible among different protein preparations. In addition, we always include controls with published indicators or fluorescent protein to confirm stable values over time, and to monitor the condition of solutions and equipments.

For experiments with neuronal cultures and brain slices, the size and type of individual samples "n" (number of dishes or slice preparations) for given experiments, is indicated and specified in the figure legends. Only for the experiment with $CdCl_2$ n refers to the total optical windows studied (two slices, on tree different positions for each one, were analyzed in this case). In the case of zebrafish experiments, for each animal all cells visible in the imaged plane (at least 4 cells/region of interest) were analyzed for the different tissues (notochord and ventral/dorsal muscle). The mean ratio GreenT-EC/mCyRFP1 per animal was then plotted. All experiments included between 4 and 6 animals per group and the precise number is indicated in the figure legends.

Statistical analyses were performed using Graphpad Prism software. Data was tested for normality using the Saphiro-Wilk test. Treated groups were compared with control animals using a two-tailed unpaired t-test to obtain the statistical significance. For multiple comparisons, One-way ANOVA was used followed by a Tukey's test. Asterisks in figures indicate $p$ values as follows: * $p < 0.05$, ** $p < 0.01$, *** $p < 0.001$ and **** $p < 0.0001$.

## Reporting summary

Further information on research design is available in the Nature Portfolio Reporting Summary linked to this article.

## Data availability

All data generated in this article can be found in the Source Data file. Mammalian and bacterial expression vectors coding for GreenT-ECs are available from Addgene (Accession Nos. 193943 to 193949). Crystal structure: The coordinates and the structure factors have been deposited in the Protein Data Bank (PDB) under the accession number: 8C0T. Source data is provided with this article. Source data are provided with this paper.

## Code availability

All python codes used for the screenings and ImageJ macros used for image analyses can be found in the Source Data file and were also uploaded to Zenodo.org (DOI: 10.5281/zenodo.8355596)[57].

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

## Acknowledgements

We thank the Protein Core Facility of the Max-Planck-Institute for Biochemistry for their assistance and the IINS Cell culture facility, especially Emeline Verdier, for primary neuronal cultures and organotypic slices. This work was supported by the Max-Planck-Society, by MISU funding from the FNRS (34772792, 40005588), from Fondation Jaumotte-Demoulin to SPS and IG, and from European Research Council (ERC-SyG ENSEMBLE, Grant No. 951294) and Human Frontiers Science Program (Grant No. RGP0036/2020) to UVN.

## Author contributions

O.G. led the study. A.V.G., U.V.N., S.S. and O.G. conceived experiments. A.V.G. conducted protein engineering, directed evolution, in vitro spectroscopy, titrations, microscopy and flow cytometry of cell lines, image processing and analysis. A.F. conducted protein evolution screenings and developed python scripts for data acquisition and analysis. I.G.S. generated transgenic zebrafish and performed in vivo confocal experiments on fish, J.A. performed experiments in rat primary neuronal cultures, A.I. and U.V.N. performed experiments in hippocampal slices of rodents, A.I. and U.V.N. analyzed experiments in hippocampal slices of rodents, J.A. and A.I. analyzed experiments in rat primary neuronal cultures, J.B. crystallized NRS F241Y, A.V.G. and O.G. wrote the manuscript, all authors contributed to discussing and revising the manuscript.

## Funding

## Competing interests

The authors declare no competing interests.
