## [Peer Review File · Nature Communications]

REVIEWER COMMENTS

Reviewer #1 (Remarks to the Author):

In the present manuscript, the authors describe the development of a series of genetically encoded fluorescent indicators for extracellular calcium. For this purpose, they inserted a minimal calcium binding domain of troponin C into mNeonGreen. Optimization led to three variants with low calcium affinity in the range of physiological extracellular calcium concentrations. The authors characterize different variants of the indicator in HEK cells, hippocampal neurons and zebrafish larvae.

Overall, the idea to generate a single-fluorophore, low-affinity calcium sensor for monitoring extracellular calcium is interesting. Such sensors enable new applications, which have not been accessible so far. However, there are several shortcomings to the manuscript that do not convince me of the suitability for publication.

First, there is only anecdotal evidence presented in several places of the manuscript and important quantifications are missing: In extended data figure 3 several strategies are presented for cell surface targeting. However, only individual examples are shown and no explanation is provided why or how the different motifs were selected. Also, it does not become clear how this figure shows "Optimization of surface delivery of GreenT-ECs". As it is, it only shows examples of seemingly randomly chosen export and membrane anchoring domains. A quantification of surface expression efficiency needs to be done.

Extended data figure 4 is limited in a similar way. Only single experiments are shown. Moreover, to substantiate the claim that indeed the intracellular localization of GreenT-EC is responsible for the low fluorescence, a time series should be shown where the membrane gets permeabilized so that intracellular calcium can reach extracellular concentrations. More generally, it is not explained why the Igk-PDGFR variant was chosen here.

Extended data figure 5 is also of limited use. No useful measures of expression and translocation are provided for the different tandem constructs. Related to this, in line 145 it is mentioned that mCyRFP was useful for the fusion construct because it "allowed convenient co-excitation of the two fluorophores with a single laser wavelength.". Are the authors referring to two-photon excitation or single-photon excitation or both? Was this the variant the authors used for the zebrafish experiments? If so, why those targeting sequences and not any other combination?

Second, the usefulness in hippocampal neurons/tissue is not clearly demonstrated. Why did the authors switch to a different form of GreenT-EC? In this case a GPI anchor is used and the identity of the signal peptide is not clear. Given the variable brightness of the fluorescence in B, it would be highly valuable to have a second, invariant fluorophore such as mCyRFP1 (as in figure 3) to test whether the brightness differences are due to local differences in calcium concentration or if they are due to the local expression of the indicator. In addition, the authors did not explain why they developed GreenT-EC.b and .c if these variants were not tested in functional experiments. It appears as if the original variant was already optimal. So, the statement in line 128 "Altogether, these mutations allowed tuning the sensors to the low affinities necessary for use in interstitial fluids, without compromising the large fluorescence changes achieved" seems exaggerated. From panel C it seems that the dynamic range of the chosen indicator is relatively limited. DF/F is less than 2fold for a change in extracellular calcium concentration likely exceeding the physiological range. This is also reflected by the relatively low change (10% DF/F) upon strong electrical stimulation. Thus, it remains unclear what physiological processes can be reliably monitored with this indicator.

My skepticisms also relates to the spatio-temporal resolution. The authors need to show a more detailed characterization of the tool. For example, how does the signal change locally? Is there any advantage of high-resolution imaging or would analysis of bulk fluorescence give the same results? What are the spatiotemporal limitations of the tool? As presented, it does not become clear how useful the indicator is for extracellular calcium imaging with decent spatial resolution. Monitoring of extracellular calcium at "high spatio-temporal resolution" as claimed in the abstract, is not shown here.

Third, the authors omit a number of significant papers, which already showed some of the

concepts presented here. For example, Zou et al. (doi: 10.1021/bi7007307) show a strategy for the insertion of a calcium binding motif into a fluorescent protein to obtain a low-affinity, single-fluorophore calcium sensor with mM affinity – very similar to the strategy presented in this manuscript. A second paper by Okkelman et al. (doi: 10.1021/acsabm.0c00649) already realized an extracellularly localized genetically encoded fluorescent calcium sensor. Also, this work is not mentioned in the present manuscript. These oversights are a bit surprising, given that that the main focus of this manuscript is on the development of a genetically encoded fluorescent sensor.

In addition to these major aspects, I have a few minor points:

- Fig. 2C: data points from individual experiments should be shown.
- Fig. 2G: why does baseline fluorescence increase?
- The image analysis described in line 417 and following cannot account for fig. 2 as it relates to ratio images
- Lines 159-160: this statement is not valid. Photobleaching is a very important aspect when using a membrane bound indicator and needs to be quantified.
- Ext. data fig. 6c: single experiments/trials need to be shown
- Ext. data fig. 7: This data snippet is not convincing. There is no correlation with massive spiking activity as opposed to the selected "ripples". Also, there is essentially no baseline shown for the middle panel. A much more rigorous quantification is needed. This includes inhibition of synaptic transmission and blockade of AP spiking in separate experiments to disentangle their contributions to extracellular calcium concentration changes and to validate that changes in brightness are indeed due to activity.
- Methods: GreenT-EC-GRAPHIC is mentioned? What does this refer to?

Reviewer #2 (Remarks to the Author):

The manuscript by Valiente-Gabioud, et al. contributes to the extensive research on developing a genetically encoded Ca²⁺ indicator (GECI). The innovative contributions of this work are: the use of directed evolution to tune the Ca²⁺ affinity of the sensor, its expression and solubility, and the anchorage to the cell membrane to target its localization, avoiding the need for high expression to overcome dilution in organism fluids. The authors characterized the properties of the GECI in vitro and in vivo, and demonstrated its space-temporal response with a resolution of a few mseconds. The lower Ca²⁺ affinities obtained allow the monitoring of the usually higher Ca²⁺ concentrations in the interstitial cellular space. As the authors stated, the developed GECI may contribute to elucidating the mechanisms that regulate the physiological levels of free calcium in body fluids and their potential compartmentalization across tissues.

The experiments are carefully performed, and the results are clearly presented.

Minor points to address:

1. If the GECI is intended to estimate interstitial Ca²⁺ concentration, it would be helpful to evaluate the correlation coefficient between the F/F₀ response and the Ca²⁺ concentration. With only two concentrations of Ca²⁺ tested, this is not possible
2. Could the authors explain why IgK-NLG shows intracellular fluorescence and EGTA insensitivity in some cells?

Typos

Remove "on" at the end of line 196 in "We incubated transgenic larvae from 2 dpf on in low calcium..."

Figure legend extended Figure 1 Line 23, "ameno acid" is misspelled; it should read amino acid

Reviewer #3 (Remarks to the Author):

The study developed a low-affinity calcium sensor that has the potential to measure interstitial calcium dynamics and regulation. While there are sensors for detecting intracellular and organelle calcium dynamics, there has been a lack of proteins that could detect changes in calcium occurring

in the mM ranges. The protein engineered in this study had an ultra-low affinity (ranging from 0.8 to 2.9 mM) for calcium, making them suitable for acting as calcium biosensors in extracellular fluids. I was excited to see that this tool was used as a proof of concept for the detection of calcium dynamics in zebrafish larvae. However, the studies were not appropriately designed to test the utility of the biosensor for a systems-level approach, which is one of the primary advantages of using zebrafish larvae. Some points to consider validating the biosensor in zebrafish:

1. It is important to show temporal changes in calcium dynamics as that is the strength of this approach. That was lacking here, and so the interpretation that the calcium levels were tightly regulated (Fig. 3 I), when exposed to different concentrations of calcium in the medium, is not insightful. I would suggest repeating the same experiments and measuring the ratio temporally in the different tissues, so we can observe the calcium dynamics in the extracellular spaces. The results will be strengthened by measuring the whole-body calcium levels in these zebrafish larvae. This would indicate the fine-tuning that is occurring in the extracellular spaces in relation to whole-body calcium homeostasis.
2. Manipulating the endogenous calcium levels to test the sensitivity of the biosensor to detect changes in extracellular calcium levels was not carried out here. One approach is to use the parathyroid hormone (PTH1) which has been shown to increase whole-body calcium levels in zebrafish larvae. This would allow the monitoring of the tissue-specific extracellular calcium dynamics temporally.
3. Fine tuning of the extracellular calcium dynamics can also be monitored by manipulating intracellular calcium dynamics using pharmacological tools (activating or inhibiting channels etc.) or by exercising the larvae with or without channel blockers.

In my opinion, the above validations are required in zebrafish larvae to fully comprehend the utility of the biosensor for detecting calcium transients against a background of high free calcium concentration.

RESPONSE TO REVIEWERS' COMMENTS

We would like to thank the reviewers for the numerous suggestions, comments and criticisms that we took as motivation to improve our manuscript. To address these, we performed new experimentation that we included in the revised manuscript, and provided corrections and clarifications, where necessary. Here is our point-by-point response to the reviewer comments.

Reviewer #1 (Remarks to the Author):

In the present manuscript, the authors describe the development of a series of genetically encoded fluorescent indicators for extracellular calcium. For this purpose, they inserted a minimal calcium binding domain of troponin C into mNeonGreen. Optimization led to three variants with low calcium affinity in the range of physiological extracellular calcium concentrations. The authors characterize different variants of the indicator in HEK cells, hippocampal neurons and zebrafish larvae.

Overall, the idea to generate a single-fluorophore, low-affinity calcium sensor for monitoring extracellular calcium is interesting. Such sensors enable new applications, which have not been accessible so far. However, there are several shortcomings to the manuscript that do not convince me of the suitability for publication.

First, there is only anecdotal evidence presented in several places of the manuscript and important quantifications are missing: In extended data figure 3 several strategies are presented for cell surface targeting. However, only individual examples are shown and no explanation is provided why or how the different motifs were selected.

Also, it does not become clear how this figure shows "Optimization of surface delivery of GreenT-ECs". As it is, it only shows examples of seemingly randomly chosen export and membrane anchoring domains. A quantification of surface expression efficiency needs to be done.

First of all, thank you for considering our approach for engineering ultralow affinity calcium biosensors interesting.

We apologize for the inaccuracy of naming the approach "optimization". In fact, it does not reflect an optimization of surface targeting motifs by dedicated systematic engineering. Instead, it involves selecting prominent combinations of signal peptides and transmembrane domains and testing them for efficient surface localization of GreenT-ECs. We corrected this in the text (revised manuscript, page 5). For quantification of surface expression efficiency, we performed flow cytometry and present the data in the new figure 3. The data show convincingly that the two most efficient surface targeting strategies use either the PDGF receptor transmembrane domain or a GPI anchor for anchoring to the plasma membrane. These are actually the targeting domains further used in the manuscript for validating GreenT-ECs.

Extended data figure 4 is limited in a similar way. Only single experiments are shown. Moreover, to substantiate the claim that indeed the intracellular localization of GreenT-EC is responsible for the low

fluorescence, a time series should be shown where the membrane gets permeabilized so that intracellular calcium can reach extracellular concentrations.

More generally, it is not explained why the Igk-PDGFR variant was chosen here.

We performed the required experiment. Supplementary figure 3b and c now shows cytosolic localization of GreenT-EC, resulting in the absence of green fluorescence, while the fused red reference protein mCyRFP1 fluoresces brightly in the red emission channel. Upon permeabilization with ionomycin green fluorescence starts to develop due to extensive calcium entry into the cytosol (Supplementary Fig. 3c). Interestingly, and completely in line with the low affinity of GreenT-EC, significant rises in cytosolic calcium appear necessary to increase GreenT-EC fluorescence. If histamine receptors are stimulated (due to activation of the G-protein coupled receptor and cytosolic release of calcium from stores), which typically raises free calcium in the cytosol of HeLa cells to a few hundred nanomolar or low micromolar concentrations, no responses can be detected with GreenT-EC, while large signals are apparent with the high affinity sensor G-CaMP6f (Fig. 3d). Thus, intracellular localization in combination with ultralow affinity and an engineered non-fluorescent apo-state of GreenT-EC are responsible for low internal fluorescence.

The Igk-PDGF receptor variant was chosen because it showed the desired degree of surface localization both in microscopy (Supplementary Fig. 2) and when quantified by flow cytometry (Fig. 3).

Extended data figure 5 is also of limited use. No useful measures of expression and translocation are provided for the different tandem constructs. Related to this, in line 145 it is mentioned that mCyRFP was useful for the fusion construct because it “allowed convenient co-excitation of the two fluorophores with a single laser wavelength.”. Are the authors referring to two-photon excitation or single-photon excitation or both?

Was this the variant the authors used for the zebrafish experiments? If so, why those targeting sequences and not any other combination?

We now quantified surface localization efficiency of the various fusion proteins using flow cytometry (Supplementary Fig. 5). The data show that fusions of GreenT-EC to mCyRFP1 are very efficiently translocated. “...convenient co-excitation of the two fluorophores with a single laser wavelength..” refers to one photon confocal microscopy (but most likely, it will also work with two photon excitation). We clarified this in the text (page 5). As pointed out already previously, the quantification of Fig 3 and the results of Supplementary Fig. 5 here indicated that GreenT-EC variants using the PDGF receptor transmembrane domain and mCyRFP1 as reference protein presented useful embodiments of the technology which we subsequently tested by transgenic expression in zebrafish.

Second, the usefulness in hippocampal neurons/tissue is not clearly demonstrated. Why did the authors switch to a different form of GreenT-EC?

In this case a GPI anchor is used and the identity of the signal peptide is not clear.

Tool development ideally should consider a range of potential application users might have in mind. As such, we were interested in presenting alternative embodiments of how GreenT-EC could be brought to use. GPI anchoring with an IgK signal peptide presented an efficient alternative for surface localization (Fig. 3). The GPI-anchored version was tested for use in neurons and overall was found to express and translocate very well. The GPI-anchored version was brought to use without reference protein. This leaves spectral bandwidth for convenient combination with other types of fluorophores to label subcellular structures, for use with red fluorescent cytosolic calcium probes or with other types of red sensors etc. Some experimenters may look for such an option.

Given the variable brightness of the fluorescence in B, it would be highly valuable to have a second, invariant fluorophore such as mCyRFP1 (as in figure 3) to test whether the brightness differences are due to local differences in calcium concentration or if they are due to the local expression of the indicator.

The differences in brightness are likely due to static differences in local expression of the sensor fluctuation of illumination intensity, and not due to different calcium levels, which would be expected to dissipate in the extracellular space. In any case, here we focus on dynamic changes of the calcium sensor, quantified as $\Delta F/F$, which indicate transient changes in calcium binding to the sensor and not changes in expression level of the sensor.

In addition, the authors did not explain why they developed GreenT-EC.b and .c if these variants were not tested in functional experiments. It appears as if the original variant was already optimal. So, the statement in line 128 "Altogether, these mutations allowed tuning the sensors to the low affinities necessary for use in interstitial fluids, without compromising the large fluorescence changes achieved" seems exaggerated.

Indeed, GreenT-EC and its affinity appear optimal for the experiments we performed. However, GreenT-ECs may be brought to use in many different types of tissues and organisms. It cannot be ruled out that other calcium concentrations will be encountered. There is, for example, the hypothesis on gradients of interstitial calcium in epidermis. There is the example of endolymphatic fluid in the cochlea, where free calcium concentrations are lower than typically found in interstitial fluids. We reasoned that experimenters might find it useful to have sensor alternatives with slightly higher and lower affinities available, should the need arise.

From panel C it seems that the dynamic range of the chosen indicator is relatively limited. $\Delta F/F$ is less than 2fold for a change in extracellular calcium concentration likely exceeding the physiological range.

We have performed more careful experiments to assess the dynamic range of the sensor both in organotypic hippocampal slices and primary neuronal cultures (Fig. 4, Supplementary Fig 6).

This is also reflected by the relatively low change (10% $\Delta F/F$) upon strong electrical stimulation. Thus, it remains unclear what physiological processes can be reliably monitored with this indicator.

According to our calibration experiments, a 10% $\Delta F/F$ change in GreenT-EC corresponds to about a 0.25 mM rise of the free calcium concentration in the ECS, which would be substantial given that the resting extracellular calcium concentration is around 1.5 mM and many physiological processes depend strongly on the extracellular calcium concentration, e.g., synaptic transmission, where the release probability of synaptic vesicles depends steeply (to the fourth power) on calcium. Moreover, we have observed comparable GreenT-EC signals for electrical stimulation spontaneous neuronal activity.

My skepticism also relates to the spatio-temporal resolution. The authors need to show a more detailed characterization of the tool. For example, how does the signal change locally?

In Figure 5b and Supplementary figure 7b we show now typical electrically induced GreenT-EC responses in neuropil and cell bodies, respectively. We did not observe any major differences among ROIs in different regions and the small fluctuations were properly averaged using a ROI for the whole image. A more exhaustive analysis comparing stratum radiatum and cell body layer confirmed this and did not show any differences in amplitude of rise and decay of the signal between these different regions.

Is there any advantage of high-resolution imaging or would analysis of bulk fluorescence give the same results?

The sensor signals were rather broad with minor regional differences in time course and amplitude. It will be interesting to do more focused experiments to map out the signals at the level of dendrites and even synapses, but this was beyond the scope of this study introducing this new tool for imaging extracellular calcium.

What are the spatiotemporal limitations of the tool? As presented, it does not become clear how useful the indicator is for extracellular calcium imaging with decent spatial resolution. Monitoring of extracellular calcium at “high spatio-temporal resolution” as claimed in the abstract, is not shown here.

We have increased the temporal resolution of our 2P imaging of GreenT-EC from 1 Hz to 7 Hz (Supplementary Fig 7c). Upon electrical stimulation, we observe that the rises in GreenT signals last several seconds as observed in the slower imaging time-lapse examples. The reasoning for that is most probably biological, i.e. the slow calcium extrusions from the cells into the ECS are happening with a second resolution. On the other hand, our acquisition parameters do not allow us to reach the temporal resolution sufficient to observe potential initial calcium coldspots that are thought to happen within tens of milliseconds.

The kinetic properties of the sensor were characterized in detail using stopped flow measurements (Fig. 2). Judging from this, the sensors have all necessary properties to detect fast and, depending on the optics employed, local transients, but the biological phenomena we encountered here are slower in nature.

Third, the authors omit a number of significant papers, which already showed some of the concepts presented here. For example, Zou et al. (doi: 10.1021/bi7007307) show a strategy for the insertion of a calcium binding motif into a fluorescent protein to obtain a low-affinity, single-fluorophore calcium sensor with mM affinity – very similar to the strategy presented in this manuscript. A second paper by Okkelman et al. (doi: 10.1021/acsabm.0c00649) already realized an extracellularly localized genetically encoded fluorescent calcium sensor. Also, this work is not mentioned in the present manuscript. These oversights are a bit surprising, given that that the main focus of this manuscript is on the development of a genetically encoded fluorescent sensor.

The main conceptual paper at the heart of our strategy is Baird et al. (1999), cited reference 17 of the manuscript. It shows that implanting a calcium binding motif into GFP renders GFP fluorescence calcium sensitive, with moderate binding affinities. All other papers describing insertion of calcium sensitive or other ligand binding domains into GFP build upon this paper. So does Zou et al. (2009). It introduces a low affinity calcium biosensor that is validated in vitro in cell lines for monitoring calcium in the endoplasmic reticulum. But we highly appreciate the work of our colleague Jenny Yang, and as the paper presents a step towards ultralow affinity calcium indicators we cite it now as part of the introduction.

The paper by Okkelman et al. does not perform genuine indicator engineering. It uses some of the FRET-based “Twitch” calcium sensors we published a while ago (Thestrup et al., 2014) and localizes them to the outside of cells by fusion to collagen-binding domains. The sensors we generated at the time are too high affinity to fit the concentrations one encounters in interstitial fluids. Also, it appears that the sensors are expressed in bacteria exogenously, purified and then added to cultures of cells, so hardly a prior art for our manuscript. But the fusion to collagen binding peptides could be interesting for bringing such sensors to work without anchoring to the surface of the plasma membrane. We thus introduced the citation in the discussion on page 9.

In addition to these major aspects, I have a few minor points:

- Fig. 2C: data points from individual experiments should be shown.

We have added the data points (now in Fig. 4 and 5)

- Fig. 2G: why does baseline fluorescence increase?

We believe that these increases were acquisition artefacts, possibly due to laser power drift, that have no biological origin. For the current version of the manuscript, we have repeated all electrical stimulation experiments, before and after 30 min of the SOV-BHH treatment, and

obtained similar responses and effects without observing this run-up (Figure 5d). Same stable behavior was observed with the rest of the new drugs tested in figure 5e-g.

- The image analysis described in line 417 and following cannot account for fig. 2 as it relates to ratio images

We have corrected for that.

- Lines 159-160: this statement is not valid. Photobleaching is a very important aspect when using a membrane bound indicator and needs to be quantified.

We now quantified the photobleaching effect of the sensor (Supplementary Fig. 7a).

- Ext. data fig. 6c: single experiments/trials need to be shown

We have corrected for that.

- Ext. data fig. 7: This data snippet is not convincing. There is no correlation with massive spiking activity as opposed to the selected "ripples".

We have drawn vertical lines between the electrophysiological recordings and GreenT-EC traces to make the temporal correlation more visible (Figure 5c and Supplementary Figure 8). The additional trace obtained in a non-stimulated slice (Figure 5c) clearly shows that spontaneous neuronal activity is correlated with GreenT-EC signal rises, providing very strong evidence that neuronal activity triggers the extrusion of calcium into the ECS.

Also, there is essentially no baseline shown for the middle panel.

Normally, we acquire nine time points of baseline before electrical stimulation. In the case of the middle panel, an event of spontaneous neuronal activity occurred during baseline acquisition, when we didn't stimulate by definition.

A much more rigorous quantification is needed. This includes inhibition of synaptic transmission and blockade of AP spiking in separate experiments to disentangle their contributions to extracellular calcium concentration changes and to validate that changes in brightness are indeed due to activity.

We have now performed several pharmacological experiments to understand better the origin of the GreenT-EC signals (new Fig. 5). Firstly, we blocked AP firing with TTX, which abolished electrically evoked GreenT-EC signals (Fig. 5e). Likewise, no spontaneous signals were observed. Secondly, we blocked voltage-gated calcium channels with cadmium chloride, which had the same effect as TTX on the GreenT-EC signals (Fig. 5f). By contrast, bath application of CNQX, which blocks postsynaptic AMPA-type glutamate receptors, had no effect on the GreenT-EC signals (Fig. 5g). Taking together, the pharmacological experiments and the observation of correlated

spontaneous GreenT-EC signals and neuronal activity provide extremely strong evidence that the changes in brightness are indeed due to activity.

- Methods: GreenT-EC-GRAPHIC is mentioned? What does this refer to?

We removed all reference to GreenT-EC-GRAPHIC.

Reviewer #2 (Remarks to the Author):

The manuscript by Valiente-Gabioud, et al. contributes to the extensive research on developing a genetically encoded Ca²⁺ indicator (GECI). The innovative contributions of this work are: the use of directed evolution to tune the Ca²⁺ affinity of the sensor, its expression and solubility, and the anchorage to the cell membrane to target its localization, avoiding the need for high expression to overcome dilution in organism fluids. The authors characterized the properties of the GECI in vitro and in vivo, and demonstrated its space-temporal response with a resolution of a few mseconds. The lower Ca²⁺ affinities obtained allow the monitoring of the usually higher Ca²⁺ concentrations in the interstitial cellular space. As the authors stated, the developed GECI may contribute to elucidating the mechanisms that regulate the physiological levels of free calcium in body fluids and their potential compartmentalization across tissues.

The experiments are carefully performed, and the results are clearly presented.

Thank you!

Minor points to address:

1. If the GECI is intended to estimate interstitial Ca²⁺ concentration, it would be helpful to evaluate the correlation coefficient between the F/F₀ response and the Ca²⁺ concentration. With only two concentrations of Ca²⁺ tested, this is not possible

The authors thank for this comment. Indeed, we have performed more careful calibration experiments with several calcium concentrations in organotypic cultures (Fig. 4g-h).

2. Could the authors explain why IgK-NLG shows intracellular fluorescence and EGTA insensitivity in some cells?

It appears that this combination of signal peptide and transmembrane domain leads to mistargeting and entrapment, potentially also to misfolding. It could be that the misfolded state is fluorescent in a calcium-independent manner. It could also be that the sensor is simply entrapped and missorted into a compartment with higher free resting calcium than in the cytosol. In either case addition of EGTA to the outside buffer would not lead to changes in fluorescence.

Typos

Remove “on” at the end of line 196 in “We incubated transgenic larvae from 2 dpf on in low calcium....”
Figure legend extended Figure 1 Line 23, “ameno acid” is misspelled; it should read amino acid

Corrected. Thank you.

Reviewer #3 (Remarks to the Author):

The study developed a low-affinity calcium sensor that has the potential to measure interstitial calcium dynamics and regulation. While there are sensors for detecting intracellular and organelle calcium dynamics, there has been a lack of proteins that could detect changes in calcium occurring in the mM ranges. The protein engineered in this study had an ultra-low affinity (ranging from 0.8 to 2.9 mM) for calcium, making them suitable for acting as calcium biosensors in extracellular fluids. I was excited to see that this tool was used as a proof of concept for the detection of calcium dynamics in zebrafish larvae.

However, the studies were not appropriately designed to test the utility of the biosensor for a systems-level approach, which is one of the primary advantages of using zebrafish larvae. Some points to consider validating the biosensor in zebrafish:

1. It is important to show temporal changes in calcium dynamics as that is the strength of this approach. That was lacking here, and so the interpretation that the calcium levels were tightly regulated (Fig. 3 I), when exposed to different concentrations of calcium in the medium, is not insightful. I would suggest repeating the same experiments and measuring the ratio temporally in the different tissues, so we can observe the calcium dynamics in the extracellular spaces. The results will be strengthened by measuring the whole-body calcium levels in these zebrafish larvae. This would indicate the fine-tuning that is occurring in the extracellular spaces in relation to whole-body calcium homeostasis.

Many thanks for the suggestions. It should be stated first that the lab in Brussels has a focus on metabolisms and homeostasis, and is not equipped for more sophisticated types of In vivo imaging. The approach they bring in, however, may be valuable to a number of laboratories as levels of interstitial calcium can be monitored by bulk tissue imaging using confocal microscopy, even in steady state, if desired. Nevertheless, we were able to show temporal changes of tissue interstitial calcium clearance after chelating calcium with EGTA (new Supplementary Fig. 9). Together with our stopped flow measurements of indicator kinetics and the temporal imaging performed in hippocampal slices this should give a good overview of what the probe could be used for. Determining whole body calcium levels in zebrafish could be indeed very interesting, but according to our information it would require several hundred fish larvae per condition to obtain reliable values. Since the focus of our work is on probe development (which has already been ongoing for about 6 years for GreenT-ECs) we considered this out of the scope of the current manuscript.

2. Manipulating the endogenous calcium levels to test the sensitivity of the biosensor to detect changes

in extracellular calcium levels was not carried out here. One approach is to use the parathyroid hormone (PTH1) which has been shown to increase whole-body calcium levels in zebrafish larvae. This would allow the monitoring of the tissue-specific extracellular calcium dynamics temporally.

We so far provided data for Calhex231, a blocker of the Calcium Sensing Receptor, which showed a significant and dose dependent decrease of interstitial calcium that was less prominent in skeletal muscle (new Fig. 6g). Applying calcitriol, the active form of vitamin D and a hypercalcemic hormone in fish, did not lead to detectable increases in interstitial calcium. But there are questions of how to apply it best, about tissue penetration or developmental periods in which an application would be most effective. We feel that we cannot deliver this within the scope of the manuscript. As new data, however, we found that Dafadine A, a CYP450 inhibitor that blocks the synthesis of the active form of Vitamin D (instead of increasing it as PTH1 would do), led to reduction in interstitial calcium levels (new Fig. 6i). This confirms the prominent role of vitamin D in regulating interstitial calcium in fish.

3. Fine tuning of the extracellular calcium dynamics can also be monitored by manipulating intracellular calcium dynamics using pharmacological tools (activating or inhibiting channels etc.) or by exercising the larvae with or without channel blockers.

In my opinion, the above validations are required in zebrafish larvae to fully comprehend the utility of the biosensor for detecting calcium transients against a background of high free calcium concentration.

Many thanks for the suggestions. Not much is currently known on the relationship and subtle interplay between cytosolic and interstitial calcium and our hope is that GreenT-ECs will stimulate numerous experiments aiming in that direction. These experiments will certainly profit from combining high resolution in vivo imaging of GreenT-EC with use of red cytosolic calcium indicators such as R-CaMPs, for example. To us they require dedicated studies in which the whole arsenal of pharmacological blockers or molecular biology tools can come to bear.

The utility of the biosensor for detecting calcium transients against a background of high free calcium is demonstrated in figure 5, where activity-evoked interstitial calcium transients are monitored against a background in free residual calcium in hippocampus. While not performed in fish, the data still demonstrate the utility of the biosensor for such purposes.

REVIEWERS' COMMENTS

Reviewer #1 (Remarks to the Author):

I thank the authors for thoroughly addressing all my critical concerns. The quality of the manuscript has significantly improved and all major issues are resolved.

I would still recommend to the authors to briefly explain in the manuscript why they chose different forms of GreenT-EC in different experiments. A short version of the explanation provided in the rebuttal letter would do. The same applies for GreenT-EC.b and .c. These explanations would provide a better orientation for the reader and make it easier to choose the appropriate tool for their application.

Simon Wiegert

Reviewer #2 (Remarks to the Author):

The authors have addressed my comments, and I see significant improvements after attending to all reviewers' comments.

Reviewer #3 (Remarks to the Author):

The authors have addressed my concerns, and I am happy for this work to be published.

RESPONSE TO REVIEWERS' COMMENTS

Reviewer #1 (Remarks to the Author):

I thank the authors for thoroughly addressing all my critical concerns. The quality of the manuscript has significantly improved and all major issues are resolved.

I would still recommend to the authors to briefly explain in the manuscript why they chose different forms of GreenT-EC in different experiments. A short version of the explanation provided in the rebuttal letter would do. The same applies for GreenT-EC.b and .c. These explanations would provide a better orientation for the reader and make it easier to choose the appropriate tool for their application.

Simon Wiegert

Many thanks for your comments. We inserted the explanation in the discussion (labeled red).

Reviewer #2 (Remarks to the Author):

The authors have addressed my comments, and I see significant improvements after attending to all reviewers' comments.

Thank you.

Reviewer #3 (Remarks to the Author):

The authors have addressed my concerns, and I am happy for this work to be published.

Thank you.